# Endogenous renal adiponectin drives gluconeogenesis through enhancing pyruvate and fatty acid utilization

Toshiharu Onodera [1], May-Yun Wang[1], Joseph M. Rutkowski[2], Stanislaw Deja [3], Shiuhwei Chen[1], Michael S. Balzer [4,5,6], Dae-Seok Kim[1], Xuenan Sun[1], Yu A. An[1,7], Bianca C. Field[1], Charlotte Lee[8], Ei-ichi Matsuo[9], Monika Mizerska[3], Ina Sanjana[9], Naoto Fujiwara[10], Christine M. Kusminski[1], Ruth Gordillo[1], Laurent Gautron [8], Denise K. Marciano[11], Ming Chang Hu [12,13], Shawn C. Burgess [3], Katalin Susztak [4], Orson W. Moe[12,13,14] & Philipp E. Scherer [1,11] ✉

Adiponectin is a secretory protein, primarily produced in adipocytes. However, low but detectable expression of adiponectin can be observed in cell types beyond adipocytes, particularly in kidney tubular cells, but its local renal role is unknown. We assessed the impact of renal adiponectin by utilizing male inducible kidney tubular cell-specific adiponectin overexpression or knockout mice. Kidney-specific adiponectin overexpression induces a doubling of phosphoenolpyruvate carboxylase expression and enhanced pyruvate-mediated glucose production, tricarboxylic acid cycle intermediates and an upregulation of fatty acid oxidation (FAO). Inhibition of FAO reduces the adiponectin-induced enhancement of glucose production, highlighting the role of FAO in the induction of renal gluconeogenesis. In contrast, mice lacking adiponectin in the kidney exhibit enhanced glucose tolerance, lower utilization and greater accumulation of lipid species. Hence, renal adiponectin is an inducer of gluconeogenesis by driving enhanced local FAO and further underlines the important systemic contribution of renal gluconeogenesis.

The mechanisms for the maintenance of serum glucose levels are essential for the supply of nutrition for the brain and other organs that utilize glucose as the principal fuel. In order to meet the constant demand for glucose, even under conditions of extremely limited caloric input, gluconeogenesis is an indispensable mechanism for survival. Glucose can be synthesized from lactate, pyruvate, glycerol and amino acids. Despite the continuous oscillations of glucose caused by gut absorption, consumption by multiple organs and endogenous production, plasma glucose levels are maintained precisely within a narrow range in healthy individuals. This accurate and highly sensitive regulation of blood glucose level is achieved by hormonal regulation, including a finely regulated insulin/glucagon ratio, neuronal regulation, glucose absorption from intestine and glucose excretion into urine.

In adult mammals, the liver, the kidney and the intestine are equipped with a full panel of gluconeogenic enzymes[1–4]. While the small intestine appears to use these pathways for lipid and amino acid trafficking[4], the proximal tubule cells of the kidney can contribute significantly to systemwide gluconeogenesis during starvation, acidosis, hypoglycemia and type 1 diabetes[5]. However, the role of renal gluconeogenesis in normal physiology and the mechanisms by which it is regulated have been vastly overlooked historically. The recent popularity of a class of blood glucose lowering drugs, the Sodium-Glucose Transporter (SGLT) 2 inhibitors, that prevent the reabsorption of glucose from urine, highlights the distinct role of the kidney in systemic glucose homeostasis by provision of glucosuria. While the liver supplies glucose into systemic circulation through a combination of glycogenolysis and gluconeogenesis, kidneys exert their vital role in

handling glucose homeostasis by a combination of gluconeogenesis and glucose excretion/reabsorption in urine.

Compared to liver gluconeogenesis that is mainly regulated by the glucagon-to-insulin ratio, kidney gluconeogenesis is more complex. The same biochemical pathways serve multiple biologic purposes. Therefore kidney gluconeogenesis can be enhanced by various factors, such as $H^+$, $Ca^+$, parathyroid hormone, catecholamines, growth hormone, glucagon, fatty acids and ketone bodies[6–11]. These factors determine the specific characteristics of renal gluconeogenesis, which are enhanced by postprandial activation of the sympathetic nervous system, acid loading and hypoglycemia. Depending on the circumstances, some of these factors can worsen hyperglycemia under conditions of diabetic ketoacidosis in type 1 diabetes and postprandial hyperglycemia in type 2 diabetes. Under normal physiological conditions, the renal gluconeogenic pathway preferentially utilizes pyruvate and glutamine as gluconeogenic substrates and the conversion of glutamine to alpha-ketoglutarate or gluconeogenesis produces two molecules of $NH_3$, which is critical to buffer urinary $H^+$. Thus, ammoniagenesis is closely linked to gluconeogenesis in the kidney.

To further complicate the situation, gluconeogenesis is energetically costly, requiring 4 ATP and 2 GTP molecules to convert pyruvate to glucose. Fatty acids and their metabolites, including acetyl coenzyme A (Acetyl CoA), citrate and ketone bodies up-regulate gluconeogenesis particularly in the renal proximal tubule[11–13]. Although fatty acids facilitate gluconeogenesis, preference for these substrates in the kidney is spatially segregated. Glucose and free fatty acids are more prone to be utilized in the medulla, while organic acids including lactate, citrate are more consumed in the cortex[14]. In other words, the renal medulla is more oxidative and glycolytic while cortex is more gluconeogenic.

Long-chain fatty acids are transported to mitochondria by carnitine palmitoyl transferase 1 (Cpt1), which facilitates their conversion to acyl-carnitine and transport prior to β oxidation. Since renal uptake of long-chain fatty acids is performed via cluster of differentiation 36 (Cd36) as a function of the fatty acid arterial concentration in vivo and in vitro[15–18], any disturbance of mitochondrial fatty acid oxidation results in lipid accumulation in renal tubular epithelial cells with potential undesired consequences.

To date, renal lipid accumulation in tubular epithelial cells has been described in several settings, including in the fasting state, obesity, aging, acute kidney injury and diabetic nephropathy as well as uric acid urolithiasis[19–26]. Although it remains to be proven whether renal lipid accumulation is directly related to the pathogenesis of each of these disease states, it is clear that dysfunctional renal fatty acid metabolism is detrimental for the tissue and leads to renal fibrosis[27].

Adiponectin is an important secreted protein, mainly produced by adipocytes in adipose tissue. Adiponectin regulates glucose and lipid metabolism and protects from atherosclerosis. Adiponectin exerts its pleiotropic beneficial effects through its receptors, adiponectin receptors 1 and 2, and T-cadherin. Elevated adiponectin levels generally reflect a high degree of metabolic health and insulin sensitivity. In fact, we have previously reported that elevated plasma adiponectin levels are potently protective in an acute kidney injury (AKI) model, whereas utilizing a global adiponectin KO model shows that these mice are slow to recover from renal injury and are potently prone to fibrosis post-AKI[28–30]. Moreover, in the clinical setting, increased serum or urinary adiponectin levels predict higher mortality in chronic kidney disease (CKD), and in the context of the nephrotic syndrome[31,32].

Adipose tissue-derived adiponectin constitutes the vast majority of serum adiponectin, but other tissues, including the kidney, heart, lung and liver can express low but locally relevant levels of adiponectin, predominantly acting in the microenvironment. In the kidney, endogenous adiponectin expression was detected in tubular epithelial cells[33]. In vitro studies demonstrate a potential anti-inflammatory role for endogenous adiponectin in the transcriptional regulation of nuclear factor kappa B (NFκB)[34]. However, the in vivo relevance of renal adiponectin remains unexplored.

In this study, we took advantage of an inducible kidney tubular cell-specific adiponectin overexpression as well as inducible kidney tubular cell-specific adiponectin knockout (KO) mice. We describe a critical role of renal adiponectin in fatty acid metabolism and gluconeogenesis with potent consequences for systemic metabolic homeostasis.

## Results

### Expression and regulation of kidney adiponectin

Compared to the high expression of adiponectin in adipocytes, adiponectin expression in other organs is limited in scope and abundance and further complicated by the fact that only a subset of cells within these alternative adiponectin-expressing tissues actually produce adiponectin.

To further explore adiponectin expression and regulation in the kidney, we compared its expression in epididymal adipose tissue (Epi), kidney, spleen, liver, intestine, brain and heart. It goes without saying that Epi expresses a high level of adiponectin given that adiponectin is the adipose tissue marker. Among these tissues other than Epi, renal expression of adiponectin is comparable to expression levels in the intestine and in the heart. Total renal adiponectin expression is 15.9 times higher than liver, but may be even higher in subsets of renal cells (Fig. 1A). The importance of adiponectin in the kidney is highlighted by significantly enlarged kidneys in adiponectin KO mice compared to control and delta-Gly adiponectin overexpressing mice at the age of 6 months, suggesting the absence of adiponectin induces kidney hypertrophy (Fig. 1B). To see this effect, the mice needed to be older, since at the age of 16 weeks, we could only observe trends towards enlarged kidneys in the adiponectin KO mice prior to providing an insult. However, upon unilateral nephrectomy (UNx), marked hypertrophy of the adiponectin KO kidneys was detected 30 days post UNx. UNx causes relatively mild kidney disease and compensatory hypertrophy of the remaining kidney by hyperfiltration[35]. This phenomenon may reflect damage or compensation caused by the loss of protection by circulating adiponectin from adipocytes, but this observation may also reflect the local action of renal adiponectin that can preserve renal function (Fig. 1C).

Adiponectin expression in human kidney positively correlates with glucagon receptor expression ($R^2 = 0.1648$, $p = 0.0013$) (Fig. 1D). This data indicates that there may be a relationship between kidney adiponectin and kidney glucagon receptor expression suggesting possibly similar regulatory mechanisms in place for renal adiponectin and renal glucagon receptor expression. This could suggest a glucoregulatory role for adiponectin as well.

Comparing the adiponectin expression between cortex and the medulla reveals that adiponectin mRNA is higher in the medulla than in the cortex (Fig. 1E). As expected, a 24-hour fast up-regulates genes that are involved in gluconeogenesis, including *phosphoenolpyruvate carboxykinase* (*Pck1*), *glucose 6-phosphatase* (*G6pase*), *pyruvate carboxylase* (*Pcx*), both in liver and in kidney. However, 24-hour fasting alone had a minimal impact on renal adiponectin expression (Fig. S1A-F). In contrast, cold exposure under fed conditions significantly up-regulated renal adiponectin expression (Fig. 1F). Moreover, the combination of 16-hour fasting and cold exposure further elevated the renal adiponectin levels as much as 9-fold (Fig. 1G). In contrast to the mRNA expression in the kidney, serum adiponectin level is lowered by cold exposure and fasting (Fig. S1G). This data suggest that the up-regulation of renal adiponectin mRNA cannot compensate the decreased level of serum adiponectin. Even though serum adiponectin is reduced, the elevated expression of renal adiponectin reflects the strong metabolic and nutritional demands under combined cold exposure and fasting.

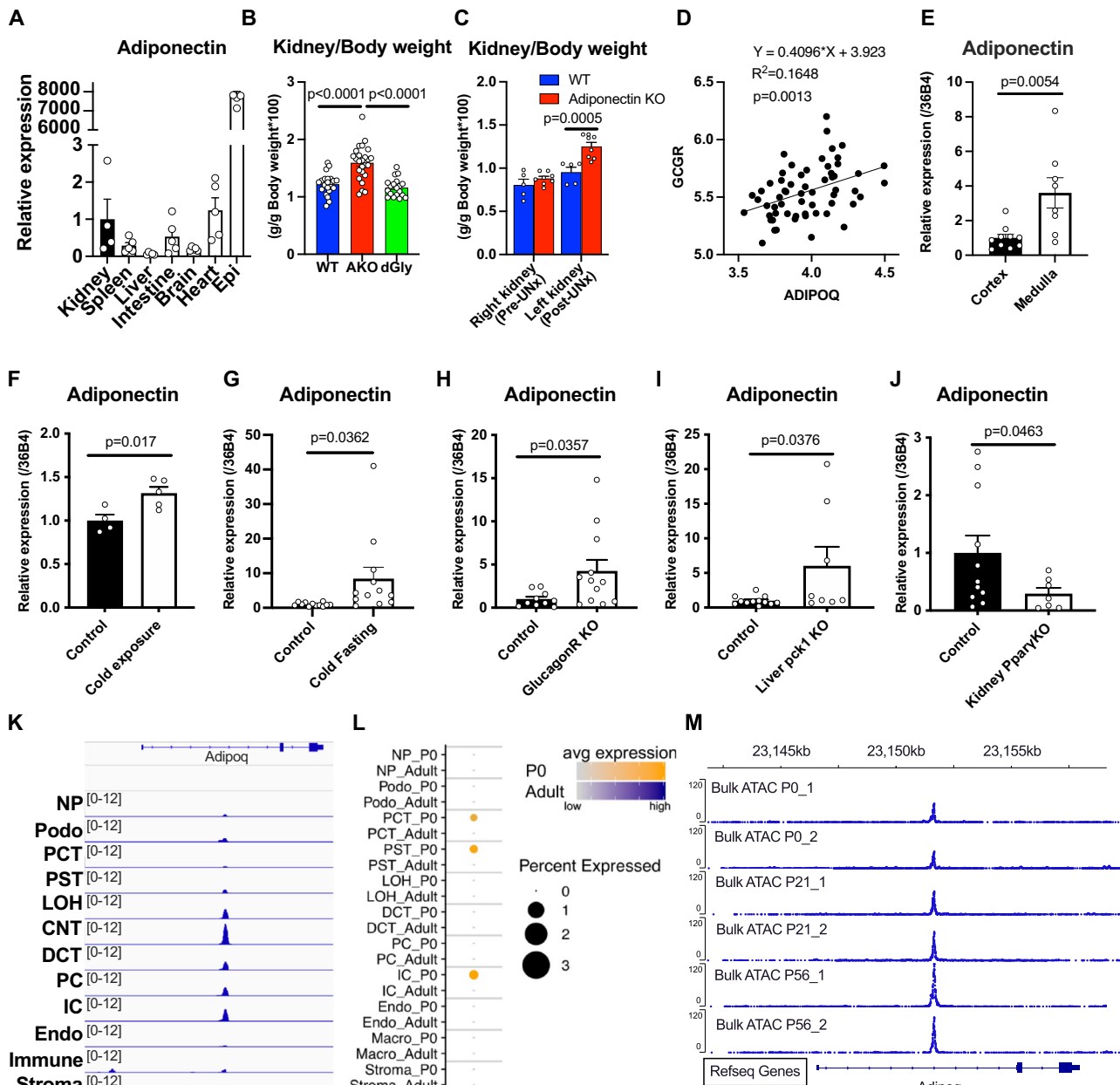

**Fig. 1 | The expression and regulation of renal adiponectin.** Determination of adiponectin expression levels, regulation and location of adiponectin expression in kidney cells using wildtype, glucagon receptor KO, liver specific Pck1 KO and kidney-specific PPARγ KO mice. **A** Adiponectin expression in Epididymal adipose tissue (Epi) (*n* = 5), kidney (*n* = 4), spleen (*n* = 5), liver (*n* = 5), intestine (*n* = 5), brain (n = 5) and heart (*n* = 5). **B** The kidney weight as a percentage of body weight in wild type (WT) (*n* = 23), adiponectin knockout (AKO) *(n* = 25) and delta-Gly adiponectin overexpression (dGly) mice (n = 19). P values were determined by Tukey's multiple comparison test. **C** The kidney weight as a percentage of body weight before and after unilateral nephrectomy (UNx) (WT n = 5, AKO n = 8) P values were determined by Sidak's multiple comparison test. **D** The expressions of ADIPOQ and GCGR in human kidney biopsy samples from Gene Expression Omnibus (n = 60). **E** Adiponectin expression in the cortex and medulla in the kidney (Cortex n = 10, medulla n = 8). **F** Adiponectin expression in the whole kidney after 1 week cold exposure under fed conditions (Control n = 4, cold exposure n = 5). **G** Adiponectin expression in the whole kidney after 3 days cold exposure and 16-hour fasting (Control and cold fasting n = 12). **H** Adiponectin expression in the kidney of global

glucagon receptor KO mice (Control *n* = 10, Glucagon R KO *n* = 12). **I** Adiponectin expression in the kidney of liver-specific Pck1 KO mice (Albumin-Cre/Pck1 flox mice) (Control *n* = 12, Liver pck1 KO *n* = 8) **J** Adiponectin expression in the kidney of kidney-specific PPARγ KO mice (Six2-Cre/PPARγ flox mice) (Control *n* = 11, Kidney PPARγ KO *n* = 7). **K** Single cell-seq reads were mapped to the reference genome. Visualization of adiponectin expression using IGV. Abbreviations are as follows. NP nephron progenitors; Podo podocyte; PCT proximal convoluted tubule; PST proximal straight tubule; LOH ascending loop of Henle; CNT connecting tubule; DCT distal convoluted tubule; PC collecting duct principal cell; IC collecting duct intercalated cell; Endo containing endothelial, vascular, and descending loop of Henle; Immune cells, including macrophages, neutrophils, lymphocytes; Stroma, stromal cell. **L** Scatter plot representation of adiponectin expression in each kidney cell type in adult kidney and postnatal day (P)0 (P0) kidneys. **M** ATAC-seq reads mapped to the adiponectin locus was visualized by IGV software. ATAC-seq was performed in duplicate utilizing whole kidney of P0, P21 and P56. Data are mean ± SEM. Unpaired two-tailed student t-tests were performed from **E** to **J**.

We utilized a global glucagon receptor KO mouse and a liver-specific Pck1 KO mouse to assess the roles of glucagon-mediated gluconeogenesis and hepatic gluconeogenic deficiency in renal adiponectin expression. Both the glucagon receptor and Pck1 are crucial for hepatic gluconeogenesis. Reports in the literature suggest that glucagon signaling is dominant for hepatic gluconeogenesis, while renal gluconeogenesis is only partially dependent on glucagon signaling[10]. While there is a positive correlation of ADIPOQ and GCGR in human kidneys, renal adiponectin is up-regulated in the global glucagon receptor KO mouse. This result raises the possibility that the up-regulation of renal adiponectin maybe a compensatory mechanism for the loss of glucagon receptor signaling (Fig. 1H). Hepatic gluconeogenic genes are down-regulated in liver Pck1 deficient mice (Fig. S2A-C), whereas the expression of gluconeogenic genes in the kidney, such as G6Pase, are significantly increased (Fig. S2D-F). Under these conditions, renal adiponectin expression is also significantly up-regulated in liver Pck1 KO mice (Fig. 1I). Results from both the glucagon receptor and hepatic Pck1 KO animals suggest that renal adiponectin is up-regulated under conditions of insufficient hepatic gluconeogenic potential. With respect to the transcriptional regulation of renal adiponectin, we analyzed the renal adiponectin expression in Six2 Cre/ PPARγ flox/flox (renal PPARγ KO) mice. PPARγ is the master regulator for the differentiation of adipocytes and governs the expression of adiponectin in adipocytes. Renal adiponectin expression is down-regulated in renal-specific PPARγ KO mice, suggesting an overlapping transcriptional regulatory network for renal adiponectin as seen in adipocytes (Fig. 1J). To further investigate the location of renal adiponectin related gene expression, single cell sequence analysis on kidneys was performed. By using an integrative genomics viewer, the aligned reads for adiponectin were visualized. The expression levels of PPARs that activate adiponectin expression are predominantly present in the kidney cortex, such as in proximal tubular cells[36], but slight expression can also be detected throughout the entire tubular component of the kidney (Fig. S3A–D). Consistent with the qPCR data, renal adiponectin expression was detected in the components that reside in the outer medulla and inner medulla, including in the proximal straight tubule (PST), in the loop of Henle (LOH), and in the cortex such as the connecting tubule (CNT), in the distal convoluted tubule (DCT) and in principal cells (PC)/intercalated cells (IC) in the collecting duct (CD) (Fig. 1K). Particularly, CNT and collecting duct cells including PC, IC and transitional cells show the highest degree of the lipid uptake-related gene expression (Fig. S3E-J). In contrast to the distribution of renal adiponectin and *Gcgr*, renal gluconeogenic enzymes, such as Pck1 and G6Pase, are predominantly expressed in proximal tubular cells (Fig. S3K-M). As another condition that renal adiponectin is induced, we also detected higher renal adiponectin at birth (P0) compared to adult mice. We detected expression of renal adiponectin in the proximal straight tubule (PST), intercalated cells (IC) and proximal convoluted cells (PCT) (Fig. 1L). These data also support the notion that adiponectin expression can be induced by a neonatal cold stimulus. We subsequently implemented an assay for transposase-accessible chromatin (ATAC) sequencing by utilizing the whole kidney at birth, postnatal day (P) 21 and P56. We could detect consistent ATAC-seq peaks for adiponectin at P0, P21 and P56 (Fig. 1M), highlighting the continuous chromatin accessibility of the adiponectin DNA locus from P0 to P56.

### RNAseq analysis of kidneys from mice with renal adiponectin overexpression reveals a gluconeogenic gene expression signature

To determine the in vivo functional role of renal adiponectin, inducible KsprtTA/TRE-Adiponectin mice (KSPAPN) were used to examine the consequences of inducible adiponectin overexpression specifically in kidney epithelial cells. Expression is induced upon administration of doxycycline. KsprtTA drives the expression of genes in the epithelia

from the proximal tubule to the collecting duct in both the adult and in the developing kidney[37] (Fig. 2A). As a first step, we performed RNAseq analysis on KSPAPN in which we overexpress adiponectin in a doxycycline-inducible manner. Hierarchical clustering of protein coding genes shows that overexpression of adiponectin in the kidney induces distinct transcriptome signatures compared to controls. Differences include 941 significantly up-regulated and 1544 significant down-regulated differentially expressed protein coding genes (Fig. 2B). Notably, among the top 10 most highly upregulated genes by adiponectin overexpression in the kidney we found 7 mitochondrial genes (Fig. 2C). Even though *Pck1* gene expression is ranked as the 13th most highly expressed gene in controls, its expression level is further doubled by the overexpression of adiponectin (Fig. 2D). A volcano plot (*p* value *vs.* fold change) identifies 81 genes up-regulated at least two-fold or more and 204 genes down-regulated to less than half (Fig. 2E). Among the up-regulated genes, we found that adiponectin overexpression increases gluconeogenesis-related genes (Fig. 2F). Taken together, these results support the idea that renal adiponectin plays an important role in promoting mitochondrial function and gluconeogenesis. qPCR analysis confirms the significant up-regulation of *Pck1* expression in the cortex of kidney (Fig. 2G). Correspondingly, increased production of Pck1 protein is detected in the cortex of the transgenic kidney (Fig. 2H), consistent with the notion that the presence of adiponectin enhances gluconeogenesis in the cortex of the kidney.

### Mice with renal overexpression of adiponectin exhibit an enhanced gluconeogenic phenotype

To corroborate the overexpression of adiponectin, we determined the renal expression of adiponectin with 1 week of doxycycline treatment. Adiponectin expression was significantly increased in the cortex and in the medulla (Fig. 3A). This is significant, as it leads to a fourfold increase in serum adiponectin protein levels, establishing that some of the locally overproduced adiponectin by the renal epithelial cells can be released into circulation at least under experimental overexpression conditions (Fig. 3B). Enhanced expression of adiponectin protein was indeed observed in kidney tissues of KSPAPN mice (Fig. 3C). KSPAPN mice were provided a high fat diet for up to 10 months to test whether adiponectin-mediated renal gluconeogenesis affects nutrient homeostasis. Systemic tolerance tests were performed after 2 months of high fat diet treatment (Fig. 3D). Bodyweight, fat mass, lean mass and tissue weights were comparable between the control and the KSPAPN group (Fig. 3E, F and G). To confirm the impact of up-regulated gluconeogenesis pathway on systemic glucose metabolism, we performed oral glucose tolerance tests (OGTTs). Although serum insulin levels during the OGTT were not altered significantly, glucose levels were increased in the KSPAPN group (Fig. 3H and I). Additionally, the KSPAPN cohort displayed insulin resistance upon exposure during an insulin tolerance test (ITT) (Fig. 3J). To determine the substrate level factors that influence glucose homeostasis in KSPAPN mice, we administered a variety of gluconeogenic precursors and oxidative substrates that provoke gluconeogenesis. We administered 3-hydroxybutyrate (3HB), which has previously been implicated as an inducer of renal gluconeogenesis[11]. 3HB increased serum glucose levels significantly, disproportionally in KSPAPN, reflecting enhanced renal gluconeogenesis by renal adiponectin overexpression (Fig. 3K). We also administered glutamine, which is thought to be an important gluconeogenic substrate for the kidney[38]. Serum glucose levels were disproportionally lower after glutamine administration in KSPAPN compared to wildtype controls (Fig. 3L). Similar to glutamine, glycerol, a third major gluconeogenic substrate for kidneys, but one that does not require Pck1, showed no effects on serum glucose levels in KSPAPN mice (Fig. 3M). In contrast, pyruvate and alanine treatment induced a significant increase in plasma glucose levels in KSPAPN mice (Fig. 3N and O), consistent

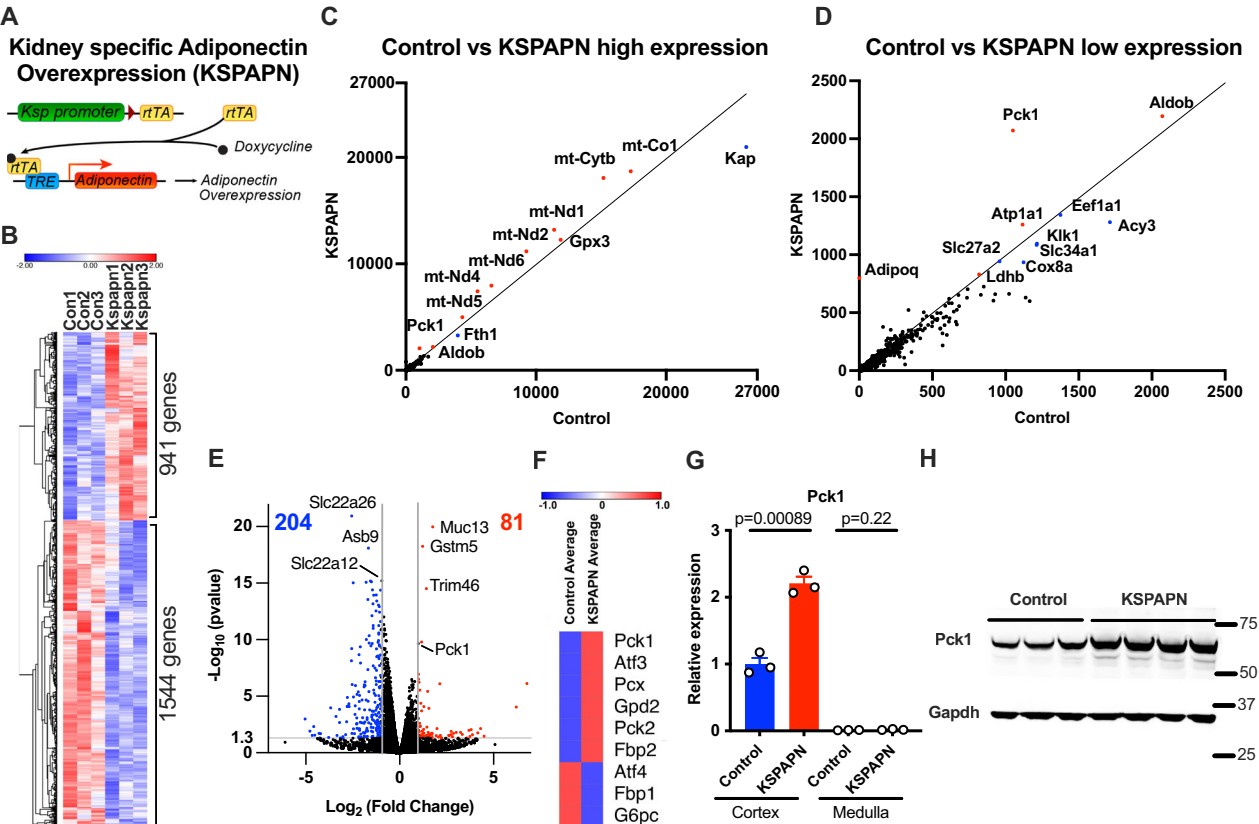

**Fig. 2 | RNAseq analysis of renal adiponectin overexpression mice exhibit higher gluconeogenic signature.** Whole kidneys were harvested after 4 days of doxycycline 600 mg/kg chow diet treatment with KsprtTA mice (Controls) and KsprtTA/TRE-adiponectin mice (KSPAPN). 3 mouse samples were pooled. 3 RNA-seq data reactions represent a total of 9 mouse samples. **A** Schematic representation of the doxycycline-inducible kidney-specific adiponectin overexpression mouse model (KSPAPN). **B** Hierarchical clustering of transcriptional profiles in control and KSPAPN ($n = 3$). **C** Scatter plot representation of protein-coding gene transcriptional profiles in controls and KSPAPN mice ($n = 3$). The range of fpkm is from 0 to 27000. **D** Scatter plot representation of protein-coding gene transcriptional profiles in controls and KSPAPN mice ($n = 3$). The range of fpkm is from 0 to 2500. **E** Volcano plots of protein-coding gene transcriptional profiles comparing *P*-value vs. fold-change. Unpaired two-tailed student t-tests were performed to determine p-value. While 81 genes were significantly up-regulated, 204 genes were significantly down-regulated. ($n = 3$). **F** Hierarchical clustering of genes involved in gluconeogenesis pathway. ($n = 3$). **G** Gene expression of Pck1 in the cortex and medulla in control and KSPAPN kidney ($n = 3$). Unpaired two-tailed student t-tests were performed to determine *p* value. **H** Western blot of Pck1 protein in control and KSPAPN kidney (Control $n = 3$, KSPAPN $n = 4$). Data are mean ± SEM.

with previous observations that pyruvate/lactate are four times better than glutamine or glycerol at provoking gluconeogenesis in rat kidney[39]. The individual data points of key tolerance tests for KSPAPN are shown in Fig. S4A-E. The fact that alanine drives higher glucose production is consistent with its rapid conversion to pyruvate. Since these experiments do not rule out hepatic gluconeogenesis as a source of increased glucose response, we tested kidney autonomous contributions in a perfused kidney model. Kidney perfusion assays with 40 mM pyruvate added to the perfusate reveal increased glucose production in KSPAPN mice (Fig. 3P). Adiponectin receptors (AdipoRs) and T-cadherin are known to act as cell surface receptors for adiponectin. AdipoR1 and 2 are ubiquitously expressed in the kidney cells. T-cadherin is mainly localized in the endothelial cells (Fig. S5A). Global AdipoR2 KO mice showed higher glucose levels during a glutamine tolerance test (Fig. S5B). Expression of genes involved in gluconeogenesis were not altered by AdipoR2 deficiency (Fig. S5C). Given that glutamine is the substrate preferentially utilized by the kidney rather than by the liver, renal AdipoR2 seems to suppress gluconeogenesis in the kidney, similar to its actions in the liver. Additionally, we assessed the renal gluconeogenesis by utilizing *ΔGly* adiponectin over-expressing mice. This mouse displays an increased *circulating* level of adiponectin in plasma, derived from adipocytes rather than the kidney. The adiponectin overexpressing mice exhibited a very diminished response in a glutamine tolerance test (Fig. S5D). This suggests that

renal intracellular adiponectin rather than circulating adiponectin is the driver for enhanced gluconeogenesis in the kidney.

Since these long term HFD treatments can induce chronic kidney injury, we proceeded to characterize overall phenotypic changes by histology and qPCR. We performed trichrome stains to evaluate the effect of local renal adiponectin expression on renal fibrosis because we have previously demonstrated potent antifibrotic effects of systemic adiponectin. The structural integrity of the kidneys was comparable between control and KSPAPN mice. Relative to wildtype kidneys, KSPAPN kidney sections exhibited a reduced trichrome stain in epithelial tubular cells and glomeruli (Fig. 3R). Gene expression analysis validated the reduced fibrotic and reduced lipogenic phenotype of the kidneys in KSPAPN mice (Fig. 3Q, S and T). In line with the results of the trichrome stain, the fibrotic cytokine *Transforming growth factor β1* (*Tgfb1*) was down-regulated by adiponectin (Fig. 3Q). Additionally, as lipogenesis contributes to the pathogenesis of fatty kidney disease[40,41], and circulating adiponectin enhances fatty acid oxidation, we examined the expression levels of genes that regulate lipogenesis. Major lipogenesis-related genes, such as *Fatty acid synthase* (*Fas*), *ATP citrate lyase* (*Acly*) and *Peroxisome proliferator receptor gamma* (*Pparγ*) were all down-regulated by renal adiponectin overexpression (Fig. 3S). Meanwhile, lipid uptake related genes, including *Cd36*, and *vesicle associated membrane protein 8* (*Vamp8*) were significantly increased by renal adiponectin overexpression (Fig. 3T).

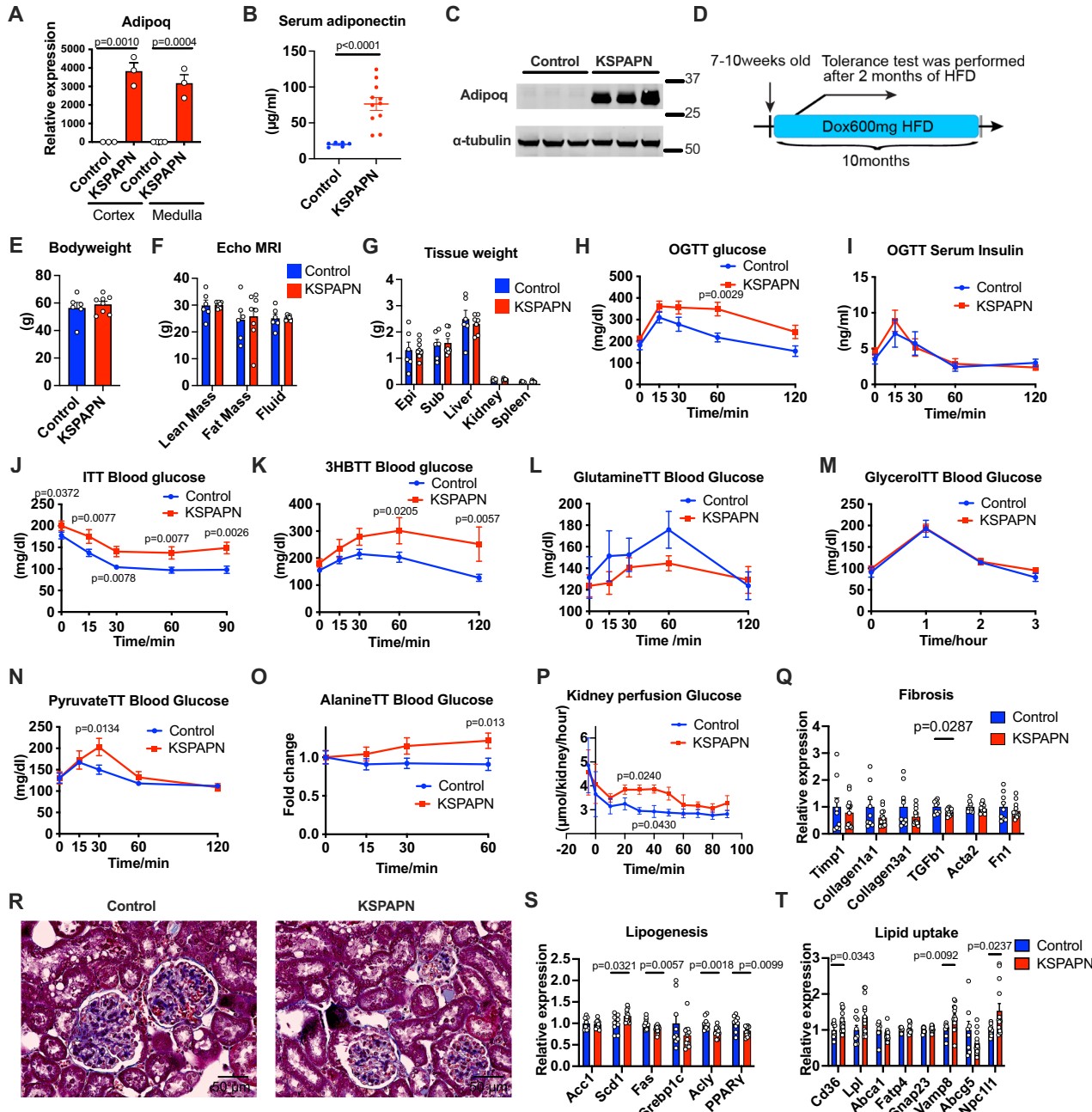

**Fig. 3 | Renal adiponectin overexpression mice exhibit an enhanced gluco-neogenic phenotype. A** Validation of adiponectin overexpression in the cortex and the medulla of the kidney after 1week doxycycline treatment (600 mg per kg diet weight) (Control Cortex $n = 3$, KSPAPN cortex $n = 3$, KSPAPN medulla $n = 3$, Control Medulla $n = 4$). **B** Serum adiponectin levels after 1week doxycycline treatment (Control $n = 6$, KSPAPN: $n = 11$). **C** Representative western blot of adiponectin (top) and α-tubulin (bottom) ($n = 3$). **D** Schematic illustration of the time course of the experiment. Doxycycline 600 mg/kg containing (dox600) HFD was started at the age of 7–10 weeks old. Tolerance tests were performed after 2months of HFD dox600. Tissues were harvested after 10 months of HFD feeding. **E** Bodyweight after 10 months of HFD dox600 (Control $n = 6$, KSPAPN $n = 8$). **F** Body composition measured by echo nuclear magnetic resonance (echo MRI) (Control $n = 6$, KSPAPN: $n = 8$). **G** Tissue weight after 10 months of HFD dox600 (Control $n = 5$, KSPAPN: $n = 10$). **H** Blood Glucose levels at different time points in OGTT (Control $n = 5$, KSPAPN: $n = 10$). **I** Blood insulin level at different time points in OGTT (Control $n = 5$, KSPAPN: $n = 10$). **J** Blood glucose level at different time points after insulin injection (Control $n = 16$, KSPAPN: n = 20). **K** Blood glucose levels at different time points after 3HB injection (Control $n = 10$, KSPAPN $n = 7$). **L** Blood glucose level at different time points after Glutamine injection (Control $n = 6$, KSPAPN $n = 11$). **M** Blood glucose levels at different time points after glycerol injection (Control $n = 6$, KSPAPN n = 11). **N** Blood glucose levels at different time points after pyruvate gavage (Control $n = 8$, KSPAPN $n = 7$). **O** Blood glucose levels (fold-change) at different time points after alanine injection (Control n = 14, KSPAPN $n = 10$). **P** Glucose production rate per kidney during the perfusion of kidneys with pyruvate-containing perfusate ($n = 3$). **Q** Gene expression of fibrosis markers (Control $n = 9$, KSPAPN $n = 13$). **R** Representative trichrome staining images of the kidney. (Scale bar: 50μm) **S** Gene expression of lipogenesis markers (Control $n = 9$, KSPAPN $n = 13$). **T** Gene expression of lipid uptake markers (Control $n = 8$, KSPAPN $n = 13$). Data are mean ± SEM. 2way ANOVA with 2-stage linear step-up procedure of BKY correction for multiple comparisons was performed to determine $p$-value from (**H**) to (**P**). Multiple unpaired two-tailed t-tests with 2-stage linear step-up procedure of BKY correction for multiple comparisons were utilized to determine $p$-values for (**Q**), (**S**) and (**T**).

Given that adiponectin overexpression increases mitochondrial gene expression, these data suggest that an increase in adiponectin facilitates the use of fatty acids for β-oxidative processes and prevents lipogenesis and fibrosis.

### Renal adiponectin alters carbons sources for circulating glucose

Since pyruvate is the primary precursor for gluconeogenesis in vivo, mice were gavaged with 40% [U-$^{13}$C$_3$]pyruvate, and its contributions to circulating and tissue metabolites were assessed (Fig. 4A). Pyruvate gavage resulted in elevated blood glucose (Fig. 4B) and blood lactate (Fig. 4C) in KSPAPN mice. In addition, lactate concentration was elevated in the KSPAPN kidney but not liver tissue (Fig. 4D), and KSPAPN mice had higher M + 3 enrichment in blood lactate and pyruvate

(Fig. 4E and F). These data indicate altered pyruvate utilization by KSPAPN mice.

To examine the fate of pyruvate during the gavage, we calculated the fractional contribution (FC) of pyruvate to various downstream metabolites. The total FC of pyruvate to downstream metabolites was calculated as the ratio of atom percent enrichment (APE) of a metabolite to the APE of circulating pyruvate. The FC specific to exogenous pyruvate was calculated as the ratio of a metabolite's APE to exogenous pyruvate APE (i.e., 40%). The difference between these values represents the FC from sources that do not enter the pyruvate pool. Although this calculation does not report metabolic rates, it provides insight into the fate of pyruvate carbons following the tracer injection. As expected, essentially all lactate was derived from pyruvate, but

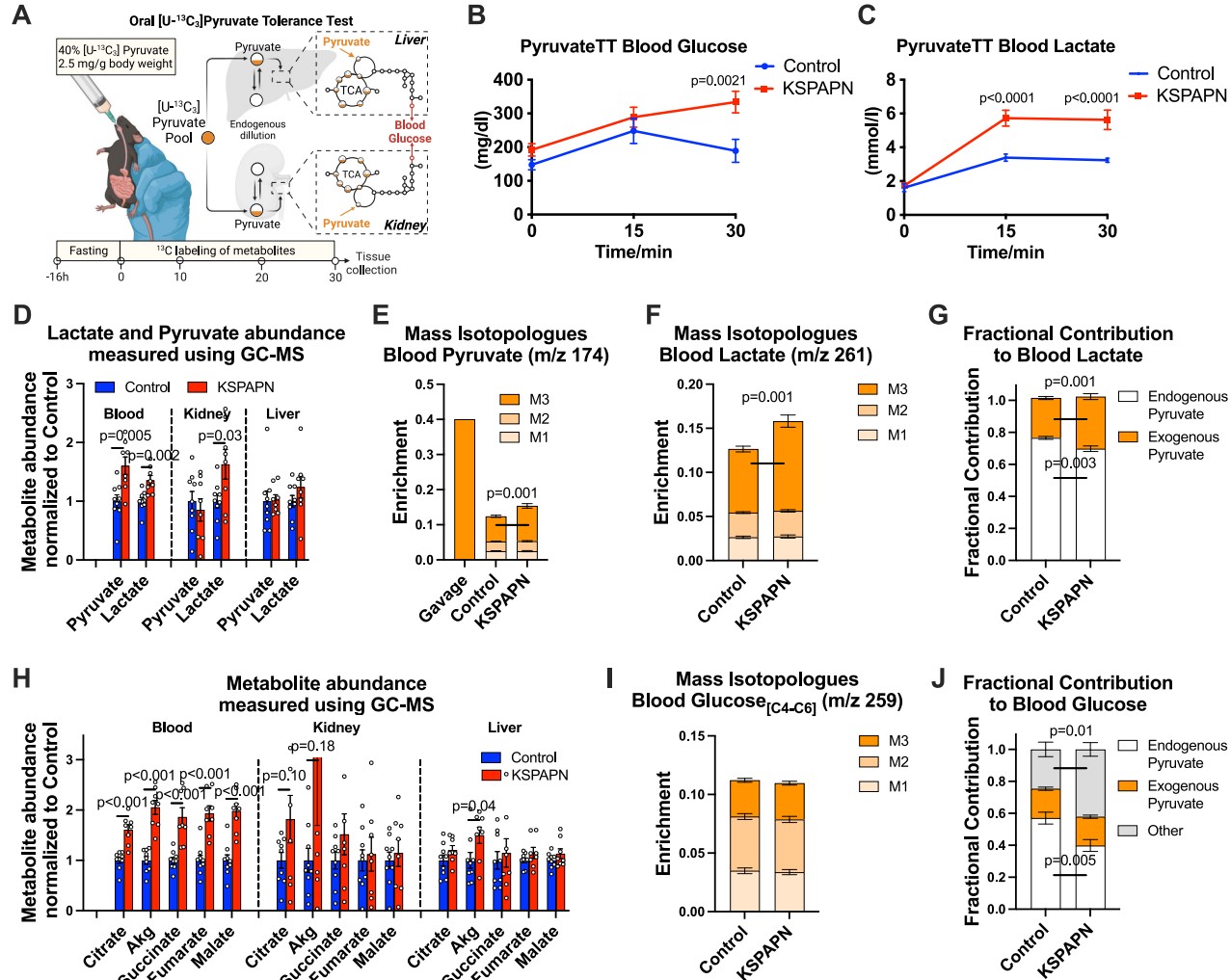

**Fig. 4 | Adiponectin affects pyruvate metabolism and sources of glucose carbon increases during OPTT. A** Schematic illustration of oral [U-$^{13}$C$_3$]pyruvate tolerance test (OPTT). 2.5 g/kg pyruvate solution containing 40% [U-$^{13}$C$_3$]pyruvate was gavaged to KSPAPN mice after 1.5 months HFD dox600 feeding. Blood, liver and kidney were harvested at the 30-minute time point. We defined the carbon contribution of pyruvate to each metabolite by analyzing the amount and percentage of each metabolite that contains labeled carbons by mass spectrometry. Created with BioRender.com. **B** Blood glucose levels at 0, 15, 30-minute time point during OPTT (Control n = 8, KSPAPN n = 8). **C** Blood lactate levels at 0, 15, 30-minute time point during OPTT (Control n = 9, KSPAPN n = 8). **D** Tissue concentration of lactate and pyruvate in KSPAPN normalized to Control at 30-minute time point (Control n = 9, KSPAPN n = 8). **E–F** Mass isotopomer distribution (MID) of (**E**) blood pyruvate (with representative 40% enrichment of gavaged

pyruvate) (Control n = 9, KSPAPN n = 8) and (**F**) blood lactate. M0 omitted for clarity of presentation (Control n = 9, KSPAPN n = 8). **G** Fractional contribution (FC) of gavaged pyruvate (40% tracer labeling in gavage solution) to blood lactate in control and KSPAPN plasma 30 minutes after the [U-$^{13}$C$_3$]pyruvate gavage FC$_{Lactate←Gavaged\ Pyruvate}$ = [APE$_{Lactate}$]/[APE$_{Gavaged\ Pyruvate}$]. (Control n = 9, KSPAPN n = 8). **H** Tissue concentration of metabolites in KSPAPN normalized to Control at 30-minute time point (Control n = 9, KSPAPN n = 8). **I** MID of blood glucose (carbons 4–6), M0 omitted for clarity of presentation (Control n = 9, KSPAPN n = 8). **J** The fractional contribution of gavaged-pyruvate, endogenous pyruvate and other carbon sources to the blood glucose (Control n = 9, KSPAPN n = 8). Data are mean ± SEM. Two-tailed unpaired Student's t test was used from (**D**) to (**J**). ANOVA with 2-stage linear step-up procedure of BKY correction for multiple comparisons was performed to determine p-values for (**B**) and (**C**).

exogenous pyruvate made a disproportionate contribution to lactate in KSPAPN mice (Fig. 4G). Nevertheless, both exogenous (originating from the bolus) and endogenous pyruvate contributed to elevated blood lactate concentration in KSPAPN mice (Fig. S6D). Interestingly, elevated blood [13]C-lactate in KSPAPN mice correlated with [13]C enrichment in kidney TCA cycle metabolites but not liver metabolites (Fig. S6F). Likewise, isotopologues of kidney lactate and citrate correlated with their blood pool equivalents, but liver metabolites had a weaker relationship, especially at higher mass shifts ($\geq M + 3$) (Fig. S6G). These data suggest plasma lactate originates from or contributes to kidney metabolites in KSPAPN mice.

Given these correlations, we examined TCA cycle intermediates following the [U-[13]C$_3$]pyruvate gavage. Plasma TCA cycle intermediates were elevated in KSPAPN mice (Fig. 4H), though these metabolites are expected to be in low concentration compared to tissue concentrations. TCA cycle intermediate concentrations tended to be higher in the kidney of KSPAPN mice, but the differences did not reach significance (Fig. 4H), and only Akg was increased in the liver of KSPAPN mice (Fig. 4H). To examine pyruvate utilization, we estimated the FC of pyruvate to TCA cycle intermediates. The FC of exogenous pyruvate to some TCA cycle intermediates was significantly higher in KSPAPN blood (Fig. S6A) and liver (e.g., malate) (Fig. S6C), but surprisingly, not in the kidney (Fig. S6B). It is important to note that the specific contribution of pyruvate to kidney TCA cycle intermediates was lower than in the liver regardless of genotype (Fig. S6B *versus* S6C). Thus, non-pyruvate anaplerotic substrates (e.g., glutamine) or higher citrate synthesis from unlabeled acetyl-CoA (e.g., via fat oxidation) relative to anaplerosis must dilute kidney TCA cycle intermediates compared to the liver. Unfortunately, the method will not detect changes in pyruvate anaplerosis in KSPAPN kidney if these pathways are systematically altered, for example by proportional increases in pyruvate anaplerosis and fat oxidation, which we address below.

Since kidney TCA cycle intermediates have a lower FC from pyruvate than liver TCA cycle intermediates, we examined blood glucose for altered contributions from pyruvate. The APE of blood glucose correlated with select liver metabolites in control but not KSPAPN mice (Fig. S6H). Despite elevated blood glucose (Fig. 4B), glucose [13]C enrichment was not significantly different in KSPAPN mice (Fig. 4I). However, since circulating pyruvate enrichment was increased in KSPAPN mice (Fig. 4E), the FC of endogenous pyruvate carbons to glucose was lower, and the FC of non-pyruvate carbon sources to glucose was higher in KSPAPN mice (Fig. 4J). Normalizing this fractional data to glucose concentration indicated that the elevated glucose excursion in the KSPAPN mice was due to increased glucose originating from exogenous pyruvate carbons and, most substantially, non-pyruvate sources (Fig. S6E). Although this method does not specifically quantify renal gluconeogenesis, the increased contribution of non-pyruvate carbons in KSPAPN mice is reminiscent of the [13]C dilution of TCA cycle intermediates in the kidney compared to liver, which may propagate to glucose more in KSPAPN mice than control mice. Nevertheless, we cannot rule out preexisting glucose due to impaired glucose clearance or increased gluconeogenesis from glycerol in KSPAPN mice, either of which would appear as non-pyruvate sources of glucose carbon.

### Renal lipid catabolism enhances renal gluconeogenesis

Since adiponectin plays a key role in triglyceride storage and metabolism in adipocytes[42,43], we conducted triglyceride tolerance tests to see whether renal adiponectin also affects systemic lipid metabolism. Serum-free fatty acid (FFA) levels, serum triglyceride and glycerol levels were not altered in KSPAPN mice during the triglyceride tolerance tests (Fig. 5A, B and C). However, β-oxidation was relatively high in the medulla and significantly increased in the cortex of the kidney under normal chow diet feeding (Fig. 5D). Consistent with the TG clearance test result, the incorporation of fatty acids was not changed

by adiponectin overexpression (Fig. S7A). Overall, the proportion of β-oxidation towards the total uptake of fatty acid was increased by adiponectin while the total uptake of fatty acid was not changed (Fig. S7B and C). In accordance with the increase in β-oxidation, the kidney tissue oxygen consumption rate (OCR) was significantly elevated, both at basal and maximal conditions (Fig. 5E and F). Along with an elevation of OCR, the extracellular acidification rate (ECAR) was significantly increased as well in KSPAPN kidneys consistent with the in vivo observation of increased blood lactate formation by kidney (Fig. 5G and H). This data also indicates that adiponectin enhances the overall metabolic rate with respect to both carbohydrate and lipid metabolism. To determine the contribution of β-oxidation to the OCR elevation, we measured OCR in the presence of Cpt-1 inhibitor etomoxir. In the presence of etomoxir, the increase of OCR by adiponectin overexpression was eliminated (Fig. 5I and J). These data strongly suggest that the elevation of OCR is attributable to an increase in β-oxidation. Consistent with these ex vivo experimental results, glucose production from the in vitro cultured kidney cells was increased by adiponectin overexpression. However, this elevation was not observed in the presence of etomoxir, suggesting that the enhanced β oxidative capacity of KSPAPN kidneys may be driving the enhanced gluconeogenic activity (Fig. 5K). It is also notable that increased fat oxidation may dilute the [13]C enrichment of kidney TCA cycle intermediates during the [U-[13]C$_3$]pyruvate gavage, thereby masking anaplerotic pyruvate utilization (Fig. S6B)

To further test whether the elevation of fatty acid catabolism by adiponectin contributes to glucose metabolism, we used additional approaches. In accordance with previous reports showing that the Cpt-1 inhibitor teglicar lowers hepatic glucose production during an OGTT[44], etomoxir decreased the blood glucose levels during an OGTT (Fig. 5L). In order to better define the contribution of β-oxidation to gluconeogenesis, we performed a lactate/pyruvate tolerance test in the presence or absence of etomoxir treatment. The differences in serum glucose levels over the course of the lactate/pyruvate tolerance tests in KSPAPN mice compared to wildtype mice were completely eliminated by etomoxir treatment (Fig. 5M and N). Taken together, this strongly suggests that the adiponectin-mediated increase in renal gluconeogenesis is completely dependent on the adiponectin-mediated local stimulation of β-oxidative capacity for fatty acids.

### Kidney-specific adiponectin KO mice exhibit impaired gluconeogenesis

To more precisely determine the function of endogenous renal adiponectin, we developed a doxycycline-inducible kidney specific adiponectin KO mouse strain (KsprtTA/TRE-Cre/adiponectin flox) ("KSPAKO"). A renal epithelial cell-specific adiponectin KO is achieved with the Ksp promoter driving the rtTA expression. Doxycycline enhances Cre expression by inducing rtTA binding to the TRE locus. Therefore, this system enables us to knockout adiponectin in a doxycycline-dependent manner in the adult mouse without any developmental issues (Fig. 6A). We fed a doxycycline-containing diet for 2 months and started to perform systemic tolerance tests (Fig. 6B). To confirm the knock down of adiponectin, we exposed KSPAKO mice to cold temperature and fasting and then performed RNAscope analysis. The adiponectin signal was significantly reduced in KSPAKO kidney (Fig. 6C). Even after 10 months of high fat diet exposure, there was no significant impact on bodyweight, fat mass or lean body mass for renal adiponectin KO mice (Fig. 6D and E). Consistent with these observations, tissue weights of KSPAKO's are all comparable to controls (Fig. 6F). In contrast to the elevation of serum adiponectin in renal adiponectin overexpressing mice, serum adiponectin levels are not altered by a lack of renal adiponectin expression (Fig. 6G). This data further corroborates that serum adiponectin is mainly derived from adipose tissues and the renal adiponectin production has little impact on levels of circulating adiponectin. Since circulating adiponectin

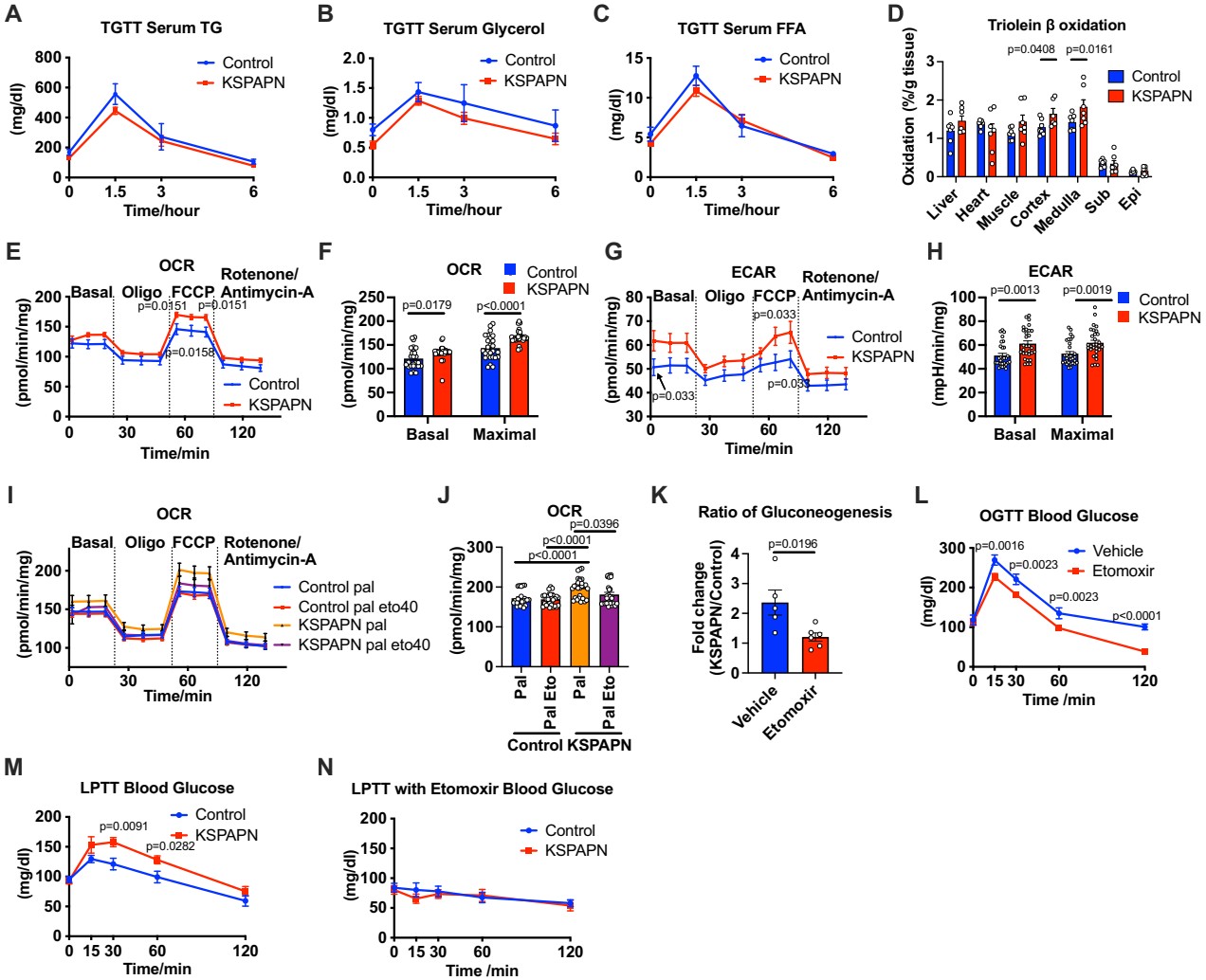

**Fig. 5 | Renal lipid metabolism enhances renal gluconeogenesis. A** Blood TG level at different time point during TG clearance test (Control $n = 6$, KSPAPN $n = 11$). **B** Blood glycerol level at different time point during TG clearance test (Control $n = 6$, KSPAPN n = 11). **C** Blood-free fatty acid level at different time points during the TG clearance test (Control $n = 6$, KSPAPN $n = 11$). **D** $^3$H-triolein lipid oxidation in tissues, including cortex and medulla, under chow diet conditions (Control $n = 7$, KSPAPN $n = 7$). Multiple unpaired two-tailed t-tests with 2-stage linear step-up procedure of BKY correction for multiple comparisons were performed to determine $p$-values **E** and **F** Oxygen consumption rate (OCR) of KSPAPN kidney tissues at basal level and after oligomycin (Oligo), FCCP and rotenone/antimycin-A treatment (Control $n = 9$, KSPAPN $n = 10$). **G** and **H** Extracellular acidification rate (ECAR) of kidney tissues at basal level, Oligo, FCCP and rotenone/antimycin-A treatment (Control $n = 9$, KSPAPN $n = 10$). **I** and **J** OCR of KSPAPN kidney tissue in the presence

of palmitate with or without etomoxir. (Control pal $n = 9$, Control pal eto40 $n = 10$, KSPAPN pal $n = 9$, KSPAPN pal eto40 $n = 9$) 1-way ANOVA with 2-stage linear step-up procedure of BKY correction for multiple comparisons were performed to determine $p$-values for (**J**). **K** The ratio of glucose production from primary cultured KSPAPN kidney cortex cells with or without etomoxir treatment (Control $n = 5$, KSPAPN $n = 6$). Unpaired two-tailed student's t-test was performed for statistics. **L** Blood glucose levels during an OGTT upon 20 mg/kg bodyweight etomoxir treatment (Vehicle $n = 8$, Etomoxir $n = 7$). **M** Blood glucose level in Lactate/pyruvate tolerance test in KSPAPN mice (Control $n = 11$, KSPAPN $n = 10$). **N** Blood glucose levels during a lactate/pyruvate tolerance test in KSPAPN mice (Control $n = 9$, KSPAPN $n = 10$). Data are mean ± SEM. 2-way ANOVA with 2-stage linear step-up procedure of BKY correction for multiple comparisons were performed to determine $p$-values for (**E**), (**F**), (**G**), (**H**), (**L**) and (**M**).

levels are unaltered in KSPAKO mice, we can specifically address the impact of renal adiponectin on kidney function. In light of our adiponectin overexpression data, we opted to further explore the gluconeogenic pathway. In contrast to the renal adiponectin overexpression, OGTT on KSPAKO mice shows lower blood glucose levels (Fig. 6H). Consistent with the improved OGTT results, an ITT reveals improved insulin sensitivity (Fig. 6I). To determine the effects of gluconeogenic substrates or ketone bodies that drive gluconeogenesis, we conducted tolerance tests with either gavage or an intraperitoneal injection of various substrates, including glutamine, pyruvate, glycerol and 3-hydroxybutyrate (3HB) and assessed the blood glucose levels over the time course. Alanine administration induced a lower production of glucose in KSPAKO compared to wild type (Fig. 6J). In contrast to renal adiponectin overexpression, the

major ketone body 3HB treatment failed to enhance glucose levels in the renal adiponectin deficient mice (Fig. 6K). Among the additional major substrates for renal gluconeogenesis, such as pyruvate, glutamine and glycerol, only pyruvate exposure leads to a lower glucose response compared to wildtype littermate controls (Fig. 6L, M and N). The individual data points of key tolerance tests for KSPAKO are shown in Fig. S4F-J. This demonstrates that renal adiponectin can enhance the utilization of pyruvate and alanine for gluconeogenesis. The fact that the serum amino acid level in KSPAPN and KSPAKO mice are not strikingly affected suggest that adiponectin-mediated renal gluconeogenesis is more related to pyruvate rather than other amino acids (Fig. S8A and B).

As adiponectin deficiency triggers fibrosis in the kidney under acute kidney injury, we evaluated the level of fibrosis after 10 months

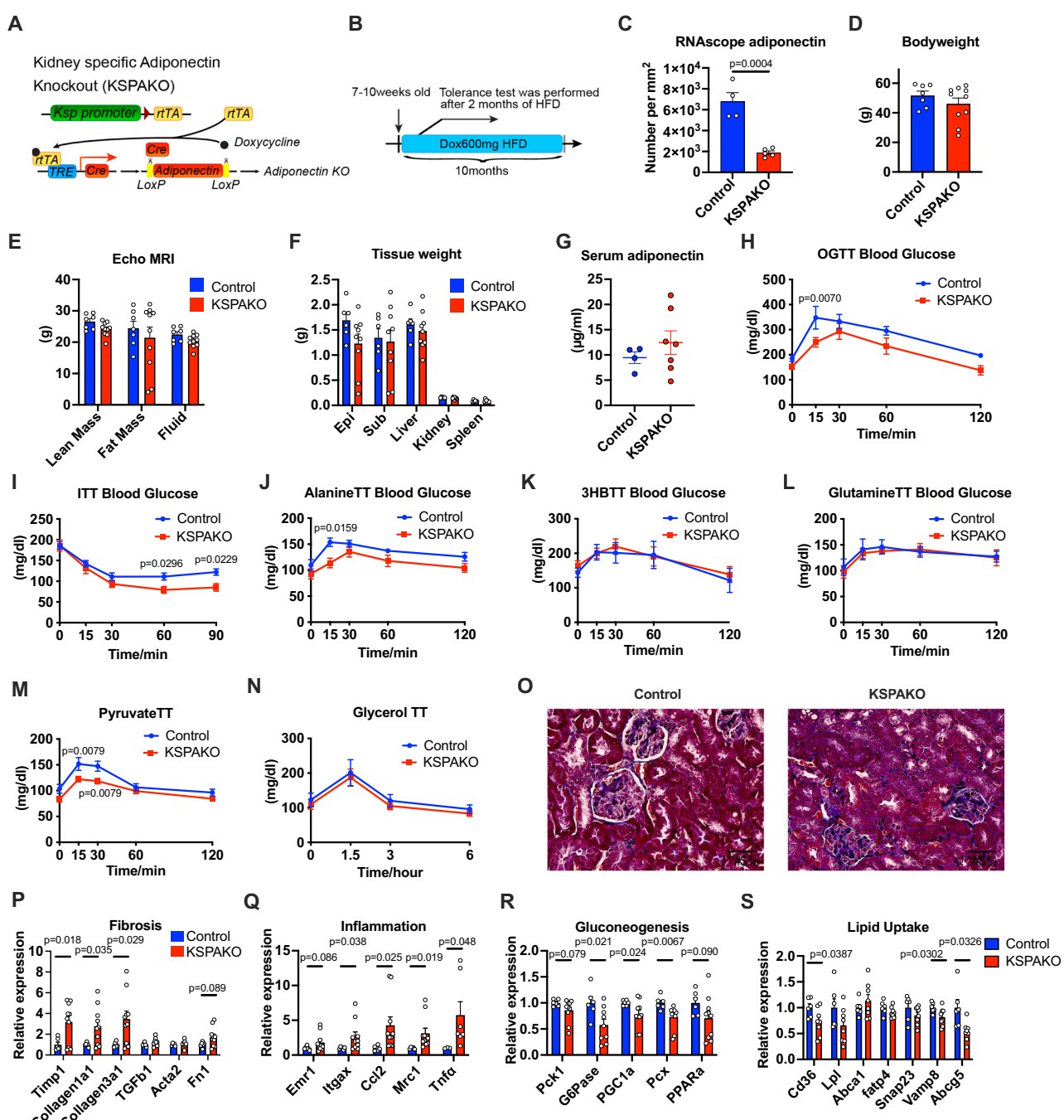

**Fig. 6 | Renal adiponectin KO mice exhibit an impaired gluconeogenic phenotype. A** Schematic illustration of the kidney-specific, doxycycline-inducible Adiponectin KO mouse model (KSPAKO). In this mouse model, KsprtTA expresses a rtTA specifically in kidney tubular cells. In the presence of Dox, rtTA activates the transcription of the TRE-Cre transgene. The Cre recombinase in turn clips the LoxP sites and converts the locus to the KO alleles. **B** Schematic representation of the time course of the experiment. Tolerance tests were performed 2 months after the start of HFD. Tissues were harvested after 10 months of HFD feeding.
**C** Quantitation of adiponectin RNAscope signal in kidney tissues per mm² (Control $n = 4$, KSPAKO $n = 5$). **D** Bodyweight of KSPAKO mice after 10 months of HFD dox600 (Control $n = 7$, KSPAKO $n = 10$). **E** Body composition measured by echo MRI (Control $n = 7$, KSPAKO $n = 10$). **F** Tissue weights of KSPAKO mice after 10 months of HFD dox600 (Control $n = 6$, KSPAKO $n = 9$). **G** Serum adiponectin levels in KSPAKO mice (Control $n = 4$, KSPAKO $n = 7$). **H** Blood Glucose levels at different time points during an OGTT (Control $n = 7$, KSPAKO $n = 9$). **I** Blood Glucose levels at different time points in ITT (Control $n = 15$, KSPAKO $n = 12$). **J** Blood glucose levels at different time points during an alanine tolerance test (Control $n = 7$, KSPAKO $n = 10$). **K** Blood glucose levels at different time points during a 3-hydroxybutyrate (3HB) tolerance

test (Control $n = 7$, KSPAKO $n = 10$). **L** Blood glucose levels at different time points during a glutamine tolerance test (Control $n = 7$, KSPAKO $n = 10$). **M** Blood glucose levels at different time points during a pyruvate tolerance test (Control $n = 10$, KSPAKO $n = 11$). **N** Blood glucose levels at different time points during a glycerol tolerance test (Control $n = 7$, KSPAKO $n = 10$). **O** Representative trichrome staining images of the kidney. (Scale bar: 50 μm). **P** Gene expression of fibrosis markers (Control $n = 6$, KSPAKO $n = 10$). (Control $n = 5$, KSPAKO $n = 10$ for Acta2). **Q** Gene expression analysis of inflammation markers (Control $n = 6$, KSPAKO $n = 10$), (Control $n = 3$, KSPAKO $n = 7$ for Tnfα), (Control $n = 5$, KSPAKO $n = 10$ for Mrc1). **R** Gene expression analysis of gluconeogenesis markers (Control $n = 6$, KSPAKO $n = 10$), (Control $n = 6$, KSPAKO $n = 10$ for G6Pase). **S** Gene expression analysis of lipid uptake markers (Control $n = 6$, KSPAKO $n = 8$). Data are mean ± SEM. 2-way ANOVA with 2-stage linear step-up procedure of BKY correction for multiple comparisons were performed to determine $p$-values for (**H**), (**I**), (**J**) and (**M**). Multiple unpaired two-tailed t-tests with 2-stage linear step-up procedure of BKY correction for multiple comparisons were performed to determine $p$-values for (**P**), (**Q**), (**R**) and (**S**).

of high fat diet. Consistent with our previous findings, kidney-specific adiponectin deficiency also exhibits a stronger tendency towards a fibrotic phenotype around the tubular epithelial cells (Fig. 6O). In fact, mRNA markers of fibrosis and inflammation-related genes are up-regulated in KSPAKO kidneys (Fig. 6P and Q). In contrast to kidney adiponectin overexpression, the expression of genes involved in gluconeogenesis are decreased in KSPAKO (Fig. 6R). In addition to the glycemic phenotype, we also found a symmetrical phenotype in lipid metabolism between KSPAPN and KSPAKO. Lipid uptake markers are generally down-regulated in KSPAKO (Fig. 6S).

### Adiponectin deficiency in renal tubular epithelial cells disrupts lipid uptake and alters lipid distribution

Adiponectin stimulates fatty acid oxidation through activation of AMP-activated protein kinase (AMPK) in myocytes[45]. Genes encoding key lipid uptake mediators are down-regulated in KSPAKO. In order to analyze the overall effect of renal adiponectin on systemic lipid metabolism, we performed a triglyceride clearance test. Renal adiponectin deletion results in impaired TG clearance from circulation upon gavage of triglycerides (Fig. 7A). A triolein uptake test reveals that incorporation of TGs is down-regulated in both the kidney cortex and medulla, explaining how impaired lipid uptake by the kidney leads to higher lipid levels in plasma over the course of the triglyceride clearance test. In contrast, the TG incorporation was elevated in the liver (Fig. 7B), which likely compensates for the decrease in lipid uptake by the kidney. Even though the total uptake of lipid was impaired by renal adiponectin deficiency, β-oxidation was not significantly affected (Fig. S7D-F). Consistent with these observations, oxygen consumption rates are decreased in KSPAKO kidneys in a Seahorse-based assay performed ex vivo (Fig. 7C and D). To explore the lipid content alteration in the KSPAKO kidneys on the basis of anatomical structure, including cortex, cortex medulla transition and medulla, we semi-quantitatively analyzed lipid species distribution considering 17117 lipid ID species at 25 μm spatial resolution through the entire kidney section using MALDI-TOF mass spectrometry imaging (Fig. 7E, F, G and H) (Supplementary Data1). By visualizing the spatial distribution alteration of $m/z$ values, we could observe a strong shift of lipid species by kidney adiponectin deficiency. As examples, $m/z$ 760.513 (putative ID as PS 34:1), $m/z$ 766.539 (putative ID as PE O-38:5;O or PE 38:4), and $m/z$ 776.498 (putative ID as SHexCer 34:2;O2) (Fig. 7F, G and H). PS 34:1 was increased throughout the entire kidney (Fig. 7F). In contrast, PE O-38:5;O/PE 38:4 was decreased in the cortex and transition area (Fig. 7G). SHExCer 34:2;O2 was increased in the cortex and medulla, but not altered in the medulla (Fig. 7H).

Because one single $m/z$ peak value in the spectrum can be matched with several lipid ID species from the database, we removed the lipid species that have multiple candidates of lipids when performing hierarchical clustering, which was conducted with 9047 species. The summation of each lipid class species revealed an overall change of the lipid content in KSPAKO and KSPAPN kidneys.

The most striking changes of lipids were observed in the transition area. To identify the tubular components of the transition area in the kidney, we performed immunofluorescence with 3 markers, including lotus tetragonolobus (LTL) (proximal tubule marker), uromodulin (Umod) (ascending limb of the Loop of Henle, straight and convoluted distal tubule marker) and calbindin1 (Calb1) (distal convoluted tubule, connecting tubule and collecting duct marker)[46]. The immunofluorescence showed that the transition area is mainly composed of LTL⁺ cells and Umod⁺ cells. This data indicate that transition area is mainly comprised of the proximal tubule, the ascending limb of the Loop of Henle and distal tubules, which partially overlaps with the distribution of endogenous adiponectin from proximal straight tubules to convoluted distal tubules (Fig. S9). In the transition area, several subtypes of sphingolipids, including hexosyl-ceramides (HexCer), acyl Ceramide (ACer), Ceramide-phosphates (CerP), Diglycerides

(DG) and Sphingomyelins (SM) exhibited a symmetrical change in KSPAKO and KSPAPN kidneys. These lipid species decreased in KSPAPN and increased in KSPAKO. In contrast, fatty acids (FA), N-acyl ethanolamines (NAE), Acyl CoAs (CoA), Phosphatidylinositol-diphosphates (PIP2), Phosphatidylinositol-triphosphates (PIP3), Bisphosphates (BMP) and Monoglycerides (MG) increased in KSPAPN and decreased in KSPAKO kidney tissues (Fig. 7I). Since MAlDI-TOF mass spectrometry imaging analyses cannot distinguish between isobaric species, we cannot exclude the possibility that not dominant species' name is annotated as a representative one among various candidates. To confirm the validity of the analysis, we analyzed the spectral data with reference to published comprehensive human and rodent kidney lipid databases whose ID have been confirmed by ultrahigh performance liquid chromatography/electrospray ionization-mass spectrometry (UHPLC/ESI-MS/MS)[47]. We can observe comparable changes of lipid species to the original global lipid database from LIPID MAPS (Fig.7I and Fig. S10). To identify the lipid species that are controlled by kidney adiponectin, we screened the lipid species that are up-regulated in the KSPAPN kidney transition area among the species down-regulated less than 0.75-fold in the KSPAKO kidneys. To avoid false positives, we limited the lipid species to those with average intensity higher than a 100 in KSPAPN transition area. The data are described as the detected $m/z$ value in negative and its corresponding matched lipid ID. Among them, $m/z$ 563.395, DG 30:4;O2 was most up-regulated in the KSPAPN transition area (Fig. 7J). On the contrary, $m/z$ 739.524, SM 32:0;O6 was the most down-regulated lipid species in KSPAPN transition area among lipid species more than 2-fold up-regulated in KSPAKO transition area (Fig. 7K).

Although we only have a limited understanding of what each of these lipid species does, we know that the accumulation of glucosylceramides is associated with renal fibrosis and mitochondrial dysfunction[48,49]. Therefore, local adiponectin reduction within the kidney has a profound impact on a broad set of lipids, including the accumulation of glucosylceramides, which may mediate decreased mitochondrial function and an enhanced fibrotic phenotype in the KSPAKO kidneys.

## Discussion

Our study aimed to elucidate the physiological relevance of adiponectin produced by kidney cells with respect to systemic glucose and lipid metabolism. While it is well-established that circulating adiponectin exerts a protective role during acute kidney injury, we lacked to date any insights into the physiologic function of kidney-derived adiponectin on systemic metabolism. Here, by taking advantage of inducible kidney-specific adiponectin gain- and loss-of-function models, we demonstrate that the local production of adiponectin in the kidney is a potent driver for fatty acid utilization from which cells derive the energy for renal gluconeogenesis.

In vivo studies focusing on specific mediators of renal gluconeogenesis have been limited in scope[50]. Importantly, the phenotypes of global Pck1 KO mice and liver-specific conditional Pck1 KO mice highlight the significance of renal gluconeogenesis to completely compensate for systemic glucose needs. Global KO of Pck1 exhibit severe hypoglycemia and die a few days after birth, while the liver specific Pck1 KO mice show normal growth and only mild hypoglycemia[51]. Recent tracer studies using multicompartment modeling confirm that renal gluconeogenesis is sufficient to compensate for loss of hepatic pck1[52]. Additionally, genetic deficiencies leading to hepatic gluconeogenic impairments, such as liver pyruvate carboxylase deficient mice, lead to an upregulation of renal gluconeogenesis[53]. This is also observed in our studies: renal gluconeogenic genes were up-regulated under fasting and cold exposure, similar to what is seen in liver Pck1 KO mice. In these settings, adiponectin was also increased in the kidney, suggesting a close relationship between adiponectin and renal gluconeogenesis. Moreover, we found

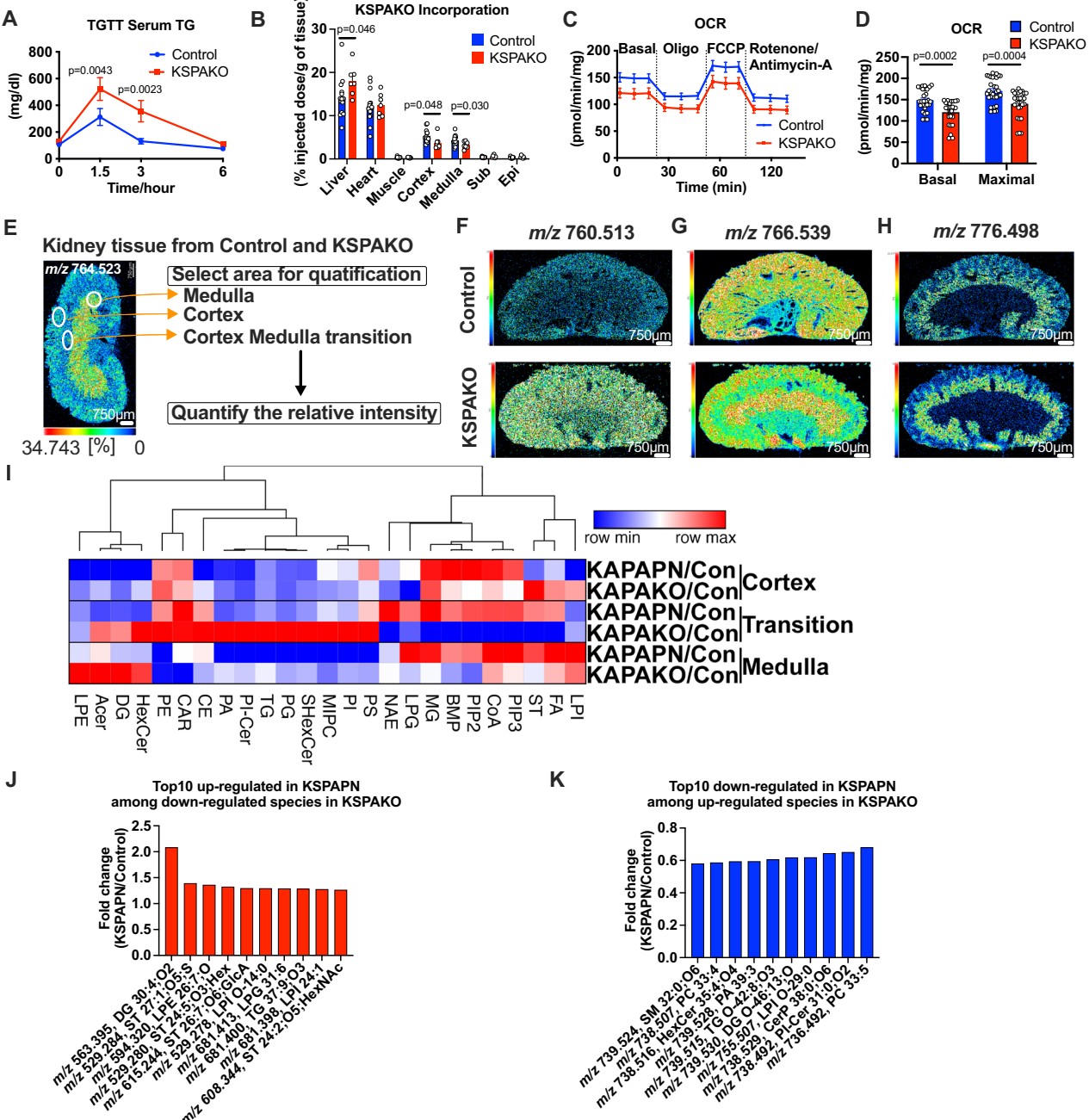

**Fig. 7 | Adiponectin deficiency in renal tubular epithelial cells disrupts lipid uptake and alters lipid distribution. A** Blood TG levels at different time points during a TG clearance test (Control $n = 8$, KSPAKO $n = 9$). **B** [3]H-triolein lipid incorporation into organs under chow diet conditions (Control $n = 14$, KSPAKO $n = 7$). Multiple unpaired two-tailed t-tests with 2-stage linear step-up procedure of BKY correction for multiple comparisons were performed to determine *p*-values. **C** and **D** OCR of KSPAKO kidney tissue at basal levels and after oligomycin (Oligo), FCCP and rotenone/antimycin-A treatment (Control $n = 10$, KSPAKO $n = 9$). **E** False color mass spectrometry image of *m/z* 764.523 depicting three main areas of the kidney; cortex, medulla and, cortex medulla transition area. For calculations, three regions of interest were selected in each main area of the kidney for each analyzed cryosection. The average intensity of each lipid specie was quantified by IMAGER-EVEAL™ MS. (Scale bar: 750 µm) **F** *m/z* 760.513 lipid specie was increased in cortex, transition area and medulla. ($n = 3$) (Scale bar: 750 µm) **G** *m/z* 766.539 lipid specie was down-regulated in the cortex and transition area. ($n = 3$) **H** *m/z* 776.498 lipid specie was increased in the cortex and transition area. ($n = 3$) **I** The heatmap of the ratio of relative intensity of lipid species (KSPAKO/control and KSPAPN/control) in the cortex, transition area and medulla. ($n = 9$) The lipid species observed at each

*m/z* value were annotated into Lyso Phosphatidylethanolamines (LPE), Acyl Ceramide (Cer) (ACer), Diglycerides (DG), Hexosyl-Cer (HexCer), Phosphatidylethanolamines (PE), Acyl carnitines (CAR), Cholesteryl ester (CE), Phosphatidic acids (PA), Phosphatidylinositol-Cer (PI-Cer), Triglycerides (TG), Phosphatidylglycerols (PG), Sulfatides (SHexCer), Mannosyl-inositolphosphoceramides (MIPC), Phosphatidylinositols (PI), Phosphatidylserines (PS), N-acyl ethanolamines (NAE), Lyso Phosphatidylglycerols (LPG), Monoglycerides (MG), Bisphosphates (BMP), Phosphatidylinositol-biphosphates (PIP2), Acyl CoAs (CoA), Phosphatidylinositol-triphosphates (PIP3), Sterols (ST), Fatty acids (FA) and Lyso Phosphatidylinositols (LPI). **J** The top10 lipid species up-regulated in KSPAPN transition area that are down-regulated less than 0.75-fold in KSPAKO transition area. **K** The top10 lipid species down-regulated in KSPAKO transition area that are upregulated more than 2-fold in KSPAKO transition area. The fold change was calculated from the relative intensity that was obtained from 3 different area in 3 of KSPAPN or KSPAKO mice. Data are mean ± SEM. 2-way ANOVA with 2-stage linear step-up procedure of BKY correction for multiple comparisons were performed to determine *p*-values for (**A**) and (**D**).

that renal adiponectin overexpression prominently increased Pck1 expression in the kidney.

Gluconeogenesis is highly coordinated with cellular energetics due to its high ATP requirements and its dependency on the TCA cycle for substrate. Thus, hepatic TCA cycle activity is closely associated with flux through pepck, which consumes TCA cycle intermediates for gluconeogenesis[54]. Fatty acid oxidation promotes gluconeogenesis through multiple mechanisms[13]. Notably, it stimulates acetyl-CoA-mediated pyruvate carboxylation, which repletes TCA cycle intermediates, and provides an ample source of ATP. As revealed by clinical studies, the kidneys consume up to 20% of the body's total energy expenditure[55] and mainly depends on β-oxidation to help meet this metabolic demand[56]. These characteristics may become more important with the activation of renal gluconeogenesis.

Adiponectin plays a pivotal role in the prevention of lipolysis from adipose tissue under insulinopenic conditions[43]. In line with this observation, kidney-specific adiponectin KO mice show lowered lipid uptake and hyperlipidemia. Additionally, kidney adiponectin overexpression leads to enhanced oxidation of fatty acids and shows higher OCR and ECAR. Thus, adiponectin energizes kidney cells through enhanced fatty acid oxidation. As a consequence, the higher ATP synthesis achieved through enhanced uptake of fatty acids and ketone bodies and subsequent fatty acid oxidation can accelerate renal gluconeogenesis[11]. In fact, ketone body administration increased blood glucose levels in KSPAPN mice, reflecting elevated renal gluconeogenesis. We also found that kidney-specific adiponectin overexpression or elimination exerts an effect on blood glucose levels following lactate/pyruvate/alanine administrations. This observation is consistent with the expected effect of fat or ketone oxidation on the activation of kidney pyruvate carboxylase[57].

Interestingly, a [U-$^{13}$C$_3$]pyruvate gavage did not produce greater $^{13}$C enrichment in kidney TCA cycle intermediates, as might have been expected with increased pyruvate carboxylase flux in KSPAPN mice. However, the method is sensitive to fractional contributions but not necessarily absolute fluxes. For example, the lack of increased $^{13}$C enrichment in TCA cycle intermediates in KSPAPN kidney may result from proportionally increased pyruvate carboxylase and β-oxidation fluxes, the former enriching TCA cycle intermediates with pyruvate carbons, and the latter diluting TCA cycle intermediates with acetyl-CoA carbons. Similarly, although a portion of the higher glucose excursion in KSPAPN mice was attributable to exogenous [U-$^{13}$C$_3$]pyruvate, the largest portion of glucose originated from non-pyruvate carbons. Since TCA cycle intermediates of the kidney were less enriched with pyruvate carbons than liver, the finding of elevated glucose originating from non-pyruvate carbons in KSPAPN mice is consistent with increased renal gluconeogenesis. Nevertheless, there are other noteworthy possibilities. Whole kidney TCA cycle intermediates may poorly reflect proximal tubule TCA cycle intermediates, where renal gluconeogenesis occurs. Indeed, the kidney normally utilizes glucose for lactate production in the medulla, while the kidney produces glucose as a whole under fasted conditions[14,58,59]. Thus the shift toward fat oxidation may lower glucose disposal in KSPAPN mice, causing pre-existing glucose to be misinterpreted as glucose derived from non-pyruvate carbon. While the entirety of the data suggest increased renal gluconeogenesis, we cannot eliminate other possibilities with respect to the in vivo tracer data.

By utilizing single cell sequencing, we found that renal adiponectin is primarily expressed from the Loop of Henle to the collecting duct. This area coincides with the area in the kidney that is highly engaged in lipid uptake-related gene expression. Interestingly, we find that renal *Gcgr* is expressed in the same region (*Wang and Scherer, unpublished*). Renal *Gcgr* affects systemic lipid levels and amino acid levels. Since serum amino acid levels were not dramatically altered by renal adiponectin deficiency or overexpression, it seems that renal adiponectin itself does not directly regulate amino acids and Gcgr.

However, the overlapping expression patterns of adiponectin and *Gcgr* suggest that they could be relevant for these pathways and mutually synergistically support their function in lipid and carbohydrate metabolism.

As renal adiponectin and Gcgr are expressed from the loop of Henle to the collecting duct, adiponectin-mediated lipid uptake should principally occur in these segments of the kidney. In contrast, gluconeogenic enzymes including Pck1 and G6Pase are principally accumulated in proximally tubular cells. This is consistent with a renal "division of labor" where lipid uptake and gluconeogenesis occur in anatomically distinct regions[14]. The observation that adiponectin leads to the induction of gluconeogenic components is not in contradiction to this model, since gene expression analysis was performed on whole kidneys and we do not have any spatial information available. Hence, the gluconeogenic intermediates stimulated by fatty acid oxidation must be exchanged between sites of lipid oxidation and sites of gluconeogenesis. Additional studies focusing on the spatially distinct sites for lipid uptake and gluconeogenesis and the exchange of energetically charges intermediates and metabolic intermediates between these sites may provide mechanistic insights into the regulation of kidney metabolism.

Adiponectin exerts its beneficial effects through its receptors AdipoR1, AdipoR2 and T-cadherin[60]. AdipoR1 and 2 were identified as cell surface receptors and utilize ceramides and AMPK as the downstream signaling molecules[61,62]. T-cadherin is also a molecule that has affinity for hexameric and multimeric adiponectin, but lacks a cytoplasmic signaling domain[63]. AdipoR1 and AdipoR2 are widely distributed in the kidney and T-cadherin is mainly expressed in kidney endothelial cells. Clinically, both adiponectin and adiponectin receptors are up-regulated in end-stage kidney disease to compensate the functional deterioration of the kidney[64]. The adiponectin receptor agonist adiporon ameriolates diabetic nephropathy[65], suggesting that signal transduction through AdipoRs prevents kidney injury. A global KO of T-cadherin shows worsened acute kidney injury, consistent with the adiponectin KO mouse phenotype[66]. In terms of renal gluconeogenesis, AdipoR2 may suppress the renal gluconeogenetic pathway, since the glucose level during glutamine tolerance tests were higher in AdipoR2 KO mice, even though the impact of AdipoR2 deficiency on renal gluconeogenic genes was limited. To analyze the impact of adipocyte derived circulating adiponectin (i.e. not derived from the kidney), we also assessed the renal gluconeogenesis in *ΔGly* adiponectin overexpressing mice[67]. Higher amounts of adipocyte-derived circulating adiponectin in *ΔGly* mice had little impact on renal gluconeogenesis, suggesting the importance of the source of adiponectin to manifest its effects on gluconeogenesis. Given that the AdipoRs are mainly localized in the plasma membrane[62], these data highlight the distinct roles of *circulating vs. intracellular* adiponectin.

The pathogenesis of chronic kidney disease and diabetic kidney disease is characterized by fibrosis and lipid accumulation[15,21]. Renal fibrosis is associated with altered lipid metabolism. Moreover, the reduced use of fatty acids and increased consumption of glucose is a predominant feature of renal cancers and polycystic kidney disease[68,69]. However, in this context, the mere increase of lipid uptake through *Cd36* overexpression in kidneys is insufficient to induce fibrosis. It is the impaired fatty acid oxidation which is more directly related to the development of renal fibrosis[27]. Consistent with this model, mice with renal adiponectin overexpression enhance fatty acid utilization and are resistant to renal fibrosis and inflammation. In contrast, kidney-specific adiponectin KO mice display reduced lipid utilization by the kidney and indeed manifest severe renal fibrosis and higher inflammation after high fat diet feeding.

Mass spectrometry-based imaging of KSPAKO and KSPAPN kidney sections allowed us to evaluate the changes of lipid accumulation with high spatial resolution. Mass spectrometry imaging provides information of relative concentration levels of molecules based on *m/z*

spectral values. Since isobaric and isomeric species cannot be resolved by MALDI-TOF imaging, further analysis by LC-MS/MS is required for unequivocal identification of the detected *m/z* features.

Nevertheless, we could observe a quantitative and distributional impact of renal adiponectin on lipid levels that would go unnoticed in a conventional analysis using whole kidney lipid extracts. Our analysis revealed that renal adiponectin deficiency triggers an up-regulation of most ceramide derivatives in the medulla and in the transition area. These data are compatible with our previous findings indicating that adiponectin deficiency leads to a build up of ceramide species[61]. Fatty acid utilization was decreased due to the lack of renal adiponectin, leading ceramide species to be further accumulated in KSPAKO kidneys, suggesting that ceramide species are building up and accumulate in the kidney. On the other hand, we discovered that fatty acids (FA), Sterols (ST), Acyl CoAs (CoA) and Phosphatidylinositol-phosphate (PIP) derivatives are up-regulated by adiponectin overexpression and decreased by adiponectin deficiency. Increased FA and CoA in KSPAPN kidneys reflect enhanced fatty acid oxidation. CoA itself displays strong bioactivity, including inhibition of critical metabolic enzymes, such as hormone-sensitive lipase[70]. Moreover, PIP derivatives are critically involved in insulin signaling. Therefore, these adiponectin-mediated alterations of lipids exert a pivotal role on kidney function.

Altogether, we demonstrate an important physiological role for renal adiponectin. It is a major player for the maintenance of serum glucose levels by pushing renal metabolism towards fatty acid utilization. Multiple lines of evidence suggest that adiponectin enhances fatty acid utilization in many different tissues. However, it seems that the response to adiponectin leading to enhanced β-oxidative capacity is tissue specific. While adiponectin suppresses gluconeogenesis in the liver by increasing insulin signaling[71], we show here that it clearly promotes gluconeogenesis in the kidney. This dichotomous role of adiponectin in hepatic *vs.* renal gluconeogenesis can be seen typically during postprandial gluconeogenesis and in hepatic insufficiency. Further studies focusing on more detailed regulatory aspects of renal gluconeogenesis will be required to gain insights as to what the basis are for these tissue-specific differences. Nevertheless, the renal gluconeogenic pathway and lipid metabolism are closely related to the pathogenesis of hypoglycemia, fat accumulation in the kidney, renal dysfunction and fibrosis. Renal adiponectin is an important player in these pathways, offering a key therapeutic target for hypoglycemia and chronic kidney disease.

# Methods

## Mice

The transgenic strains *KsprtTA/TRE-Adiponectin mice* (KSPAPN), *adiponectin flox mice* and *ΔGly adiponectin overexpression mice* were generated by our laboratory as previously described[43,67,72]. Kidney-tubular cell specific adiponectin knockout mice (KSPAKO) were generated by crossing *KsprtTA/TRE-Cre mice* and *adiponectin flox mice*. The *PPARγ flox mouse* was purchased from the Jackson Laboratory (RRID:IMSR_JAX:004584)[73]. *Glucagon receptor KO mice* were provided by Dr. Maureen Charron at Albert Einstein College of Medicine[74]. *Liver specific Pck1 KO mice* were generated by Dr. Mark Magnuson, Vanderbilt University[51]. Global *AdipoR2 KO mice* were generated by Deltagen, Inc. All mice were bred in the C57BL/6 genetic background. Mice were fed on a regular diet (LabDiet #5058) or high-fat diet (60% energy from fat, Research Diets #D12492). Mice were maintained under barrier conditions in 12-h dark/light cycles in a temperature-controlled environment (22 °C), with *ad libitum* access to diet and water and constant veterinary supervision. Water and cages were autoclaved and cages were changed every other week. The humidity is monitored and controlled from 30 to 70%. Male mice were utilized for this study. All protocols for mouse use and euthanasia were reviewed and approved by the Institutional Animal Care and Use Committee of the University

of Texas Southwestern Medical Center, animal protocol number 2015-101207G.

## Genotyping PCR

Approximately 3 mm of mouse tail tip was incubated in 80 μL 50 mM NaOH at 95 °C for 1.5 h. 8 μL 1 M Tris–HCl (pH 8.0) was added for neutralization. After vortexing and a short spin down, 0.5–1 μL of supernatant was used as PCR template. Primer sequences for genotyping PCR are listed in Table S1. The PCR program was: 95 °C for 5 min, followed by 35 cycles of 95 °C for 15 s, 62 °C for 30 s, and 72 °C for 30 s, and ended with 72 °C for 3 min.

## Immunohistochemistry

Mice were euthanized by cervical dislocation under isoflurane anesthesia. Tissues were immediately collected and fixed in 10% buffered formalin for 24 hours. Afterwards, tissues were rinsed with 50% ethanol and embedded in paraffin blocks and sliced for 5 μm sections. Trichrome staining was performed according to the manufacturer's instructions (#ab150686, Abcam).

## Assay of metabolites

Tail blood was collected with a Microvette CB300Z (SARSTEDT #16.440.100) for serum and Microvette CB300 K2E (SARSTEDT #16.444.100) was used for plasma. Glucose was measured with a glucose meter or colorimetric assays with PGO enzymes (Sigma #P7119) plus o-dianisidine (Sigma #F5803). Insulin was measured with an ELISA kit (Crystal Chem #90080).

## qPCR analysis

Total RNA was extracted with an RNeasy Mini Kit (Qiagen #74106) for pancreas or Trizol (Invitrogen #15596018) for kidney, liver, adipose tissue and other organ. cDNA was synthesized with iScript cDNA Synthesis Kit (Bio-Rad #170-8891). Quantitative real-time PCR (qPCR) was performed with the Powerup SYBR Green PCR Master Mix (Applied Biosystems # A25742) on Quantistudio 6 Flex Real-Time PCR System (Applied Biosystems # 4485694). Primer sequences for qPCR are listed in Table S2.

## Western blotting

Protein was extracted from kidney tissue by homogenization in PBS supplemented with 1 mM EDTA, 20 mM NaF, 2 mM $Na_3VO_4$, and protease inhibitor cocktail. 5× RIPA buffer was added to the homogenate for a final concentration of 10 mM Tris-HCl, 2 mM EDTA, 0.3% NP40, 0.3% deoxycholate, 0.1% SDS, and 140 mM NaCl, pH 7.4. The sample was centrifuged at $10,000\,g$ for 5 minutes. 20–50 μg/lane of supernatant protein was separated by SDS-PAGE (NP0335BOX, Thermo Fisher) and transferred to nitrocellulose membrane. The blots were then incubated overnight at 4 °C with rabbit anti-mouse polyclonal PCK1 antibodies (ab70358, Abcam) in a 1% BSA TBST-blocking solution. Primary antibodies were detected using secondary antibodies labelled with infrared dyes emitting at 700 nm or 800 nm (1:5000, Li-Cor Bioscience 925-68073 and 926-32213, respectively) The Odyssey Infrared Imager was used to visualize Western blots with Li-Cor IRdye secondary antibodies.

## Systemic assays

For oral glucose tolerance test (OGTT), mice were fasted for 4–6 h and subjected to an oral gavage of dextrose (2 mg/g body weight). Tail blood was collected at 0, 15, 30, 60, and 120 min and prepared for serum and assayed for glucose and insulin. For insulin tolerance test (ITT), insulin (1.5 U/kg Humulin R; Eli Lilly, Indianapolis, IN, USA) was administered under fed condition. Serum glucose level was measured at 0, 15, 30, 60, and 120 minutes time points. Triglyceride tolerance test (TGTT) was initiated by oral gavage of 20% Intralipid (10 μl/g BDW, l141-100mL, Sigma), and serum was collected at 0, 1.5, 3, and 6 hr for

triglyceride, NEFA and glycerol assay. Glucose, insulin and triglyceride levels were measured using Contour blood glucose monitor (9545 C, Bayer) or an oxidase-peroxidase assay (Sigma P7119), insulin ELISA (Crystal Chem, Elk Grove village, IL, USA, #90080) and Infinity Triglycerides Reagent (Thermo Fisher Scientific TR22421), respectively. Glycerol and NEFA were measured by free glycerol reagent (F6428, Sigma) and free fatty acid quantification kits (Wako Diagnostics-NEFA-HR2), respectively. Blood lactate level was measured by lactate plus (Nova Biomedical). For 3-hydroxybutyrate, glutamine, glycerol and alanine tolerance test, substrate was administrated (2.5 mg/g body weight) respectively by IP injection after overnight fasting. The pyruvate tolerance test (PTT) was performed by oral gavage of pyruvate (2.5 mg/g body weight) after overnight fasting. Etomoxir (20 mg/kg bodyweight) was given by intraperitoneal (IP) injection 30 minutes prior to the initiation of substrate administration.

## Mitochondrial respiration measurements

Kidney tissue oxygen consumption rate was determined using the Seahorse XFe24 Analyzer (Agilent) following the manufacturer-recommended BOFA (basal-oligomycin-FCCP-antimycin A/rotenone) protocol. Harvested kidneys were cut into half longitudinally, then punched out by 2 mm biopsy punch (33-31, Integra Miltex). Approximately 4 mg of kidney tissues were utilized for Seahorse measurements. For kidney specific adiponectin overexpressing kidney tissues, the assay buffer was composed of 1 mM pyruvate with XF base medium (102353-100, Agilent) instead of 1 mM pyruvate, 2 mM glutamine and 7 mM glucose. Ex vivo mitochondrial function was measured by utilizing 3–4 mg kidney tissue. For tissues, oligomycin (10 μM), FCCP (8 μM) and antimycin A (4 μM) plus rotenone (2 μM) were added during the assay. OCR and extracellular acidification rate (ECAR) were recorded through the Seahorse instrument. For determining the fatty acid utilization, palmitate (100 μM, 29558, Cayman) and etomoxir (40 μM, 11969, Cayman) were originally dissolved in the media.

## Triolein uptake assay

$^3$H-triolein (#NET431001MC; PerkinElmer, Waltham, MA) was retro-orbitally intravenously injected (2 μCi per mouse in 100 μl of 5% intralipid) into mice after a 16 h fasting. Blood samples (0.15 ml) were then collected at 1, 2, 5, 10 and 15 min after injection. At 15 min following injection, mice were euthanized, blood samples were taken and selective tissues were harvested. Tissues were quickly excised, weighed and frozen in liquid nitrogen and stored at −80 °C until processing. Lipids were then extracted using a chloroform-to-methanol based extraction method[75]. The radioactivity content of tissues, including blood samples, was quantified by scintillation counter (Tri-Carb 2910 TR, PerkinElmer, Waltham, MA).

## Body composition analysis

The measurements of mouse whole-body composition including total water, total fat mass and lean mass were performed by the EchoMRI whole body composition analyser (Echo Medical Systems, Houston, Texas, USA).

## Unilateral nephrectomy

Removal of the right kidney was performed after tying the renal pedicle and ureter with a 6−0 nylon suture, followed by post operative analgesia. The weight of right kidney was measured as a control. The left kidney weight was measured and harvested 30 days after nephrectomy.

## Primary kidney cell isolation

Whole kidneys were collected and capsules were removed. The cortex of the kidney was sliced and minced into small pieces followed by the incubation in digestion buffer (HBSS, 0.1 mg/ml trypsin inhibitor T6522, Sigma, 1 mg/ml collagenase #C0130, Sigma) at 37 °C for 45 min.

After digestion, the mixture was passed through 70-μm cell strainers and then centrifuged at 500x*g* for 5 min. Cell pellets were resuspend in DMEM containing 10%FBS and seeded in the cell culture dishes. After reaching confluency, kidney primary cells were passaged to appropriate dishes for the experiments.

## In vitro gluconeogenesis assays

Prior to each glucose production assay, cells were starved with DMEM containing no glucose, no pyruvate (A14430-01, Gibco) but supplemented with 5 mM HEPES and 10 nM Dexamethasone and penicillin / streptomycin (basal media) for 4 h. Kidney cells were seeded in a 96-well plate for this assay. For glucose production assays, the basal media was replaced with the basal media supplemented with 9 mM lactate, 1 mM pyruvate, 2 mM alanine, 2 mM glutamine, 50 μM palmitate, 10 μM forskolin and with or without 40 μM etomoxir. After 24-hour incubation, glucose level was measured by colorimetric assays with PGO enzymes (Sigma #P7119). Glucose production was normalized by the kidney cell protein content.

## RNA-seq

RNA-sequencing was performed by Novogene (Sacramento, CA, USA) by utilizing isolated RNA from the control kidney and adiponectin overexpressed kidney. After the QC procedures, mRNA from eukaryotic organisms is enriched from total RNA using oligo(dT) beads. For prokaryotic samples, rRNA is removed using a specialized kit that leaves the mRNA. The mRNA from either eukaryotic or prokaryotic sources is then fragmented randomly in a fragmentation buffer, followed by cDNA synthesis using random hexamers and reverse transcriptase. After first-strand synthesis, a custom second-strand synthesis buffer (Illumina) is added, with dNTPs, RNase H and Escherichia coli polymerase I to generate the second strand by nick-translation and AMPure XP beads are used to purify the cDNA. The final cDNA library is ready after a round of purification, terminal repair, A-tailing, ligation of sequencing adapters, size selection and PCR enrichment. Library concentration was first quantified using a Qubit 2.0 fluorometer (Life Technologies), and then diluted to 1 ng/ml before checking the insert size on an Agilent 2100 and quantifying to greater accuracy by quantitative PCR (Q-PCR) (library activity >2 nM). Libraries are fed into Novaseq6000 machines according to activity and expected data volume.

## Oral [U-$^{13}$C$_3$] Pyruvate Tolerance Test

Control and KSPAPN mice were used for metabolic tracing after 1.5 months of HFD and dox600 feeding. Overnight fasted mice were gavaged with 2.5 mg/g of a 40% [U-$^{13}$C$_3$] Pyruvate solution. Blood glucose and lactate were monitored at 0, 15 and 30 minutes post gavage. At the 30-minute time point, mice were anesthetized using isoflurane and blood, liver and kidney tissues were harvested and immediately snap frozen in liquid nitrogen.

## Metabolite sample preparation for GC-MS

Blood glucose enrichment analysis was conducted as described previously[76]. Briefly, 25 μL of plasma was deproteinized using an excess of cold acetone (12:1 v/v), vortexed, centrifuged (4 °C, 21000×g, 10 min) and supernatant decanted. The procedure was repeated with 25 μL of MiliQ H$_2$O. Supernatants were combined and dried under air. Glucose was converted to aldonitrile pentapropionate (ALDO) by adding 50 μL of 2% hydroxylamine hydrochloride and incubating at 90 °C for 60 min. Next, 100 μL of propionic anhydride was added, and samples were incubated at 60 °C for 30 min. The solvent was evaporated under air, and the samples were dissolved in 100 μL of ethyl acetate, centrifuged (4 °C, 21000×g, 10 min) and transferred into GC vials containing a glass insert.

Tissue samples (~40–60 mg) were homogenized with 600 μl of ice-cold 80% MeOH and 20 μL of a norleucine standard. Samples were

incubated for 10 minutes on ice and then centrifuged (4 °C, 21000×g, 15 min). The supernatant was transferred and dried under $N_2(g)$. Next, 50 μL of 1% metoxylamine hydrochloride in pyridine was added to the samples and incubated for 90 min at 37 °C. Then, 80 μL of a silylation reagent, MTBSTFA, was added and samples were incubated at 60 °C for 60 min. The derivatives were transferred into GC vials containing a glass insert and analyzed in SIM mode.

## Metabolite enrichment analysis and semi-quantitative analysis using GC-MS

Metabolite analysis was performed using an Agilent 7890-A GC-MS system equipped with an HP-5ms column (30 m × 0.25 mm I.D., 0.25 μm film thickness; Agilent J&W) combined with an Agilent 5975-C mass spectrometer (70 eV, electron ionization source). For all samples, a 1 μL injection volume was used and split mode was adjusted for optimal signal-to-noise ratio. For analysis of the ALDO derivative, the following temperature gradient was used: 80 °C for 1 min, ramped at 20 °C/min to 280 °C, held for 4 min and ramped at 40 °C/min to 325 °C. After a 5 min solvent delay, MS data collection was initiated and scan range was set to 170–380 $m/z$. For analysis of organic acid enrichment, the following temperature gradient was used: 120 °C for 2 min, increased 6 °C/min to 210 °C, increased at 25 °C/min to 250 °C and held for 1 min, and finally increased by 25 °C/min to 275 °C and held for 7 min. After a 3.9 min solvent delay, MS data collection was initiated. SIM ranges were as follows: pyruvate (174.1–178.1 $m/z$), lactate (261.1–265.1 $m/z$), alanine (260.1-264.1 $m/z$), norleucine (200.1 $m/z$), succinate (289.1–294.1 $m/z$), fumarate (287.2–292.2 $m/z$), α-ketoglutarate (AKG, 346.2–352.2 $m/z$), malate (419.2–424.2 $m/z$), aspartate (418.2-423.2 $m/z$), glutamate (432.3-437.3 $m/z$), phosphoenolpyruvate (PEP, 453.3-458.3 $m/z$), citrate (431.2-438.2 m/z) and 459.2–466.2, and 3-phosphoglycerate(3PG, 585.4-590.4 $m/z$). Metabolite signal areas were integrated using MassHunter software. For semi-quantitative analysis, metabolite signal areas (consisting of the sum of all isotopologue abundances) were normalized to the area of internal standard (norleucine) divided by tissue weight used for analysis. All mass isotopomer distributions (MIDs) were corrected for natural abundance in INCA software[77], based on the theoretical isotopic distributions of individual chemical fragment formulas.

## Relative metabolite contribution estimations

Labelling of a metabolite in the blood or tissue resulting from $^{13}C$ isotope-tracer was expressed by calculating atom percent excess (APE):

$$APE_{Metabolite} = \sum_{i=1}^{n} \frac{M_i \times i}{n} \qquad (1)$$

where $i$ iterates the number of possible $^{13}C$-labeled carbons in the metabolite, $n$ is the total number of potentially labeled carbons in the metabolite, and $M_i$ is the fractional abundance of the $i$th isotopologue.

The fractional contribution (FC) of [U-$^{13}C_3$]pyruvate tracer to tissue and blood metabolites ($FC_{Metabolite \leftarrow Pyruvate}$) was determined by dividing their respective APE:[78]

$$FC_{Metabolite \leftarrow Pyruvate} = \frac{APE_{Metabolite}}{APE_{BloodPyruvate}} \qquad (2)$$

An index of the total blood glucose originating from various sources was calculated as the FC multiplied by the concentration of the product metabolite concentration (i.e. blood lactate or blood glucose). Contributions of pyruvate to glucose were as follows:

Blood glucose from gavaged pyruvate tracer:

$$FC_{BloodGlucose \leftarrow GavagedPyruvateTracer} = \frac{APE_{BloodGlucose\,C4-C6}}{APE_{PyruvateTracer}} \times Blood\,Glucose\left[\frac{mg}{dL}\right] \qquad (3)$$

Blood glucose from all (endogenous and gavaged) pyruvate associated gluconeogenesis:

$$FC_{BloodGlucose \leftarrow BloodPyruvate} = \frac{APE_{BloodGlucose\,C4-C6}}{APE_{BloodPyruvate}} \times Blood\,Glucose\left[\frac{mg}{dL}\right] \qquad (4)$$

Blood glucose from non-pyruvate associated gluconeogenesis:

$$FC_{BloodGlucose \leftarrow Non-Pyruvate} = 1 - FC_{BloodGlucose \leftarrow BloodPyruvate} \qquad (5)$$

## Kidney perfusions

Mice were anesthetized with 2% isoflurane in 95% oxygen/5%carbon-dioxide air gas. A midline incision to the peritoneal cavity was made, and the major abdominal blood vessels were exposed. Two tied ligatures were placed around the celiac artery and mesenteric artery. One ligature was placed around inferior vena cava at the position right below right renal vein junction, into which heparin (100 units per mouse) was injected. After that, a ligature was placed at the proximal end of the aorta right below the mesenteric artery junction. An open ligature was placed around the aorta right above the celiac artery and tied off after an 18 G blunt end needle was installed for buffer infusion. The perfusion pressure was maintained below 80 mm of mercury at the rate 0.8-1 ml per minute. The buffer temperature was adjusted around 37 °C by an in-line heater (Warner Instrument). Basic composition of the buffer is as follows (mM). NaCl, 129; Hepes, 20; KCl, 4.8; Bicarbonate, 5; $CaCl_2$, 2.4; $MgSO_4$, 1; $KH_2PO_4$, 1; no glucose; pyruvate, 40; BSA, 0.65%; Dextran, 3.6%. The perfusates were collected at 10-min intervals for 100 min. Glucose levels in perfusates were measured by colorimetric assays with PGO enzymes (Sigma #P7119).

## RNAscope

RNAscope multiplex fluorescent assays were performed by the UT Southwestern Metabolic Phenotyping Core to detect the transcripts for our genes of interest in fixed frozen section of kidneys. Slides were pretreated following the manufacturer's instructions (Advanced Cell Diagnostics, Inc (ACD), California). After boiling slides in a Target Retrieval solution, slides were rinsed in distilled water and dehydrated in 100% ethanol. Tissue was enzymatically digested at 40 °C for 15 mins. Hybridization itself was performed following the recommended ACD procedure and reagents from the RNAscope® multiplex Detection Kit V2 (cat# 323110). The probe (Mm-Adipoq-O2-C1 cat# 1244281) were applied at 40 °C for 2hrs. Amplification steps were done following the manufacturer's instructions using Opal dyes 520 and 570. Slides were counterstained with DAPI before applying mounting medium (ProLong™ Gold Antifade Cat# P36930) and a coverslip over the tissue section.

## Bioinformatic analysis

Differential expression of genes between control and KSPAPN were analyzed by using genes with an fpkm of ≥0 in all samples. To generate a heatmap, significantly changed protein coding genes were extracted from the original fpkm expression data. Hierarchical clustering was performed after normalization based on Z-score by Morpheus (https://software.broadinstitute.org/morpheus/). The scatter plot and volcano plot were generated by GraphPad Prism (GraphPad, San Diego, Calif., USA).

## Single cell RNA sequencing of P0 and adult mice

Single cell RNA sequencing was performed in the Dr. Susztak laboratory. One-day-old wild type mouse neonate and adult mouse (C57BL/6) kidneys were harvested and minced into 1 mm³ pieces and incubated with digestion solution containing Enzyme D, Enzyme R, and Enzyme A from the Multi Tissue Dissociation Kit 1 (Miltenyi, 130-110-201) at 37 °C for 15 min with agitation. The reaction was deactivated by adding FBS to 10%, then the solution was passed through a 40 μm cell strainer. After centrifugation at $500 \times g$ for 5 min, the cell pellet was incubated with 500 μL of RBC lysis buffer on ice for 3 min. We centrifuged the cells at $500 \times g$ for 5 min at 4 °C and resuspended the cells in 1X PBS for further steps. Cell number and viability were analyzed using Countess AutoCounter (Invitrogen, C10227). The cell concentration was 2.2 million cells/mL with 92% viability. Ten thousand cells were loaded into the Chromium Controller (10X Genomics, PN-120223) on a Chromium Single Cell B Chip (10X Genomics, PN-120262), and processed to generate single cell gel beads in the emulsion (GEM) according to the manufacturer's protocol (10X Genomics, CG000183). The library was generated using the Chromium Single Cell 3′ Reagent Kits v3 (10X Genomics, PN-1000092) and Chromium i7 Multiplex Kit (10X Genomics, PN-120262) according to the manufacturer's manual. Quality control for constructed library was performed with the Agilent Bioanalyzer High Sensitivity DNA Kit (Agilent Technologies, 5067-4626) for qualitative analysis. Quantification analysis was performed with the Illumina Library Quantification Kit (KAPA Biosystems, KK4824). The library was sequenced on an Illumina HiSeq 4000 system with $2 \times 150$ paired-end kits using the following read length: 28 bp Read1 for cell barcode and UMI, 8 bp I7 index for sample index and 91 bp Read2 for transcript.

## Bulk ATAC sequencing

**Nuclei preparation.** Bulk ATAC sequencing was performed in Susztak laboratory with previously published methods[79,80]. Kidneys were minced and lysed in 5 mL lysis buffer (10 mM Tris HCl pH 7.4, 10 mM NaCl, 3 mM MgCl₂, and 0.1% Nonidet™ P40 Substitute in nuclease-free water) for 15 min. The reaction was then blocked by adding 10 mL 1× PBS into each tube, and solution was passed through a 40 μm cell strainer. The nuclei were centrifuged down at $500 \times g$ at 4 °C, resuspended in the resuspension buffer (10 mM pH = 7.5 Tris-HCl, 10 mM NaCl, 3 mM MgCl₂). Nuclei numbers were estimated with Countess AutoCounter (Invitrogen, C10227).

**Transposition.** Fifty thousand nuclei/sample were tagmented with Tagment DNA TDE1 Enzyme and Buffer Kit (Illumina, 20034198) in 50 μL reaction volume of transposition buffer (25 μL 2× TD buffer (Tagment DNA Buffer), 2.5 μL Tn5 transposase (Tagment DNA Enzyme 1), 0.5 μL 10% Tween-20, 0.5 μL 1% Digitonin, 16.5 μL 1× PBS, 5 μL nuclease-free water). The reaction was carried out at 37 °C for 30 min in a thermomixer.

**DNA purification and library construction.** Isolated DNA was purified with the MinElute Reaction Cleanup Kit (Qiagen, 28204) by following the manufacturer's manual. The purified DNA was finally eluted in 10 μL elution buffer. The DNA was then amplified by NebNext High-Fidelity 2× PCR Master Mix (NEB, M0541S) and quantified by qPCR to make libraries. The libraries were purified with AMPure XP beads (Beckman Coulter, A63880). Quality control for constructed library was performed with the Agilent Bioanalyzer High Sensitivity DNA kit. Libraries were submitted to 150 bp PE sequencing with Illumina HiSeq 3000 system.

## Amino acid measurements

Plasma amino acids profiling was performed at the UTSW Metabolic Phenotyping Core. Plasma samples were collected from 47-week-old male mice housed at room temperature under *ad libitum* conditions.

Plasma samples were thawed on ice. One microliter of EDTA free protease inhibitors cocktail was added to 100 μL of plasma (1 tablet of cOmplete™ EDTA-free protease Inhibitor Cocktail, (Cat# 5056489001 Millipore-Sigma, Bedford, MA), was dissolved in 1 mL of saline solution). Samples were processed and analyzed in a CLAM 2030 fully automated sample preparation module for LCMS coupled to a Nexera X2 UHPLC system and an LCMS-8060 triple quadrupole mass spectrometer (Shimadzu Scientific Instruments, Columbia, MD). The sample preparation sequence was programmed as follows: addition of 60 μL of MeOH onto CLAM 2030 filtration vial; addition of 10 μL of sample; addition of 10 μL of internal standard cocktail; vortexing for 90 seconds at 1,900 rpm; addition of 50 μL of MeOH; vortexing for 60 seconds at 1,900 rpm, filtration for 70 seconds. The sample collected into the collection vial was then automatically injected for LC/MS/MS analysis (1 μL injection). Free amino acids were analyzed using the mass spectrometry parameters and chromatographic conditions described in the Shimadzu LC/MS/MS Method Package for Cell Culture Profiling. The method was edited to include stable isotope labeled free amino acids internal standards SRM transitions. LabSolutions V 5.82 and LabSolutions Insight V 2.0 program packages were used for free ammino acids mass spectrometry data processing (Shimadzu Scientific Instruments)[81].

## Renal gene expression analysis in clinical samples

The whole transcriptome data of human kidney biopsy samples from chronic kidney disease patients were obtained from the Gene Expression Omnibus (https://www.ncbi.nlm.nih.gov/geo/). Kidney biopsy samples were obtained from the European Renal cDNA Bank (ERCB) cohort and microarray analysis was performed by Grayson et al. [82]. (GSE104954-GPL22945). The GSE104954-GPL22945 Data set includes 7 diabetic nephropathy, 7 Focal segmental glomerulosclerosis, 4 minimal change, 21 ANCA-associated vasculitis, 3 tumor nephrectomy and 18 unannotated samples.

## Mass Spectrometry (MS) Imaging. MALDI TOF Lipids Imaging

Kidney tissues from control, KSPAPN and KSPAKO after 2 weeks of high fat diet (HFD) containing doxycycline 600 mg/kg (dox600) were harvested and frozen in the vapor of liquid nitrogen. Frozen kidney tissues were sectioned at 10 μm on a Leica CM3050 S Cryostat (Leica Microsystems) at object temperature of −20 °C and chamber temperature of −20 °C. Sections were thaw-mounted onto indium tin oxide coated glass slides (Millipore-Sigma, 578274) and stored at −80 °C until analysis. 9-aminoacridine (9-AA) was used as a MALDI matrix and was applied using an automated matrix vapor deposition system iMLayer™ (Shimadzu Corporation, Kyoto, Japan) using a sublimation method at a matrix film thickness of 0.9 μm.

An iMScope™ QT MALDI Q-TOF mass spectrometer (Shimadzu Corporation) equipped with an optical microscope and an Nd:YAG laser was used for acquiring mass spectrometry imaging (MSI) data. MALDI MS spectra were acquired in the $m/z$ range of $m/z$ 500 to 1,000 in the negative ion mode. The scanning pitch was set to 25 μm using the instrumental settings for the laser diameter setting value of 2 and the laser intensity setting values of 74. The other parameters were set as follows: sample voltage, 3.70 kV; detector voltage, 2.50 kV; number of laser shots, 125; laser repetition rate, 5000 Hz. Optical and MSI data were processed by IMAGEREVEAL™ MS software (Shimadzu Corporation). 3 pairs of control and KSPAKO kidneys were prepared. 3 regions of interest in each cortex, medulla and cortex medulla transition area were selected. In total, 9 regions in each kidney structure were quantified for analysis. Putative identification of lipids was performed ($m/z$, 5 ppm tolerance) via cross-referencing with the LIPID MAPS database (The LIPID MAPS® Lipidomics Gateway, https://www.lipidmaps.org/) using IMAGEREVEAL™ software. $m/z$ values matching multiple lipid IDs, were removed for the calculation of heatmap (Fig. 7l). The heatmap includes the species that exhibited p value < 0.10 (Control vs

KSPAKO in transition area). Multiple lipid candidates were listed when several IDs were matched with the same *m/z* values (Fig. 7J and Fig. 7K).

## Statistical analysis

All data were expressed as mean ± SEM. Differences between two groups were examined for statistical significance by the Student's t-test. One-way Analysis of variance test (ANOVA) with Tukey's test or 2way ANOVA with a two-stage linear step-up procedure of Benjamini, Krieger, and Yekutieli (BKY) by controlling the False Discovery Rate (<0.05) using Prism software (GraphPad, San Diego, Calif., USA) was applied to the multiple comparisons. *P* value or adjusted P value < 0.05 denoted the presence of a statistically significant difference. Specific statistics used in each figure are described in the figure legends.

## Reporting summary

Further information on research design is available in the Nature Portfolio Reporting Summary linked to this article.

## Data availability

All data supporting the findings of this study are available within the article and its supplementary materials, including Source Data. Source data are provided with this paper. RNA sequencing data are available at NCBI GEO under the accession code (GSE242095). Raw Imaging mass spectrometry data is available in METASPACE:(https://metaspace2020.eu/project/kidney_adipoq_ko) (https://metaspace2020.eu/project/kidney_adipoq_oe). Source data are provided with this paper.

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

## Acknowledgements

We thank the UTSW Animal Resource Center, Histology Core, Metabolic Phenotyping Core, the Live Cell Imaging Core, Transgenic Core and Flow Cytometry Facility for their excellent assistance with experiments performed here. We also thank Shimadzu Scientific Instruments for the collaborative efforts in mass spectrometry technology resources. This study was supported by US NIH grants RC2-DK118620, R01-DK55758, R01-DK099110, R01-DK127274 and R01-DK131537 to P.E.S. K01-DK133630 to SD. R01-DK078184 and Atkins Foundation to SCB. R01-DK081423 and R01-DK115703 to OM. We would also like to acknowledge support of the UTSWNORC grant under NIDDK/NIH award number P30-DK127984.

## Author contributions

T.O. designed and performed the experiments and acquired data with the help of M.Y.W., R.G., S.C., Y.A., B.F., C.L., X.S. and P.E.S.. T.O. and P.E.S. interpreted the data and wrote the manuscript. C.M.K. helped the manuscript preparation. D.K. and N.F. assisted in RNAseq analysis. M.S.B performed ATAC-seq and single cell sequence. T.O., S.D. and M.M. performed tracer experiments. L.G. performed RNAscope. E.M. and I.S. assisted in the MS Imaging analysis. D.M., O.M., K.S., S.B., M.C.H., J.M.R., provided feedback on the manuscript. P.E.S. is the guarantor of this work and, as such, had full access to all the data in the study and takes responsibility for the integrity of the data and the accuracy of the analysis.

## Competing interests

The authors declare no competing interests.

## Additional information

**Peer review information** : *Nature Communications* thanks Ton Rabelink and the other, anonymous, reviewer(s) for their contribution to the peer review of this work. A peer review file is available.

[1]Touchstone Diabetes Center, The University of Texas Southwestern Medical Center, Dallas, US. [2]Division of Lymphatic Biology, Department of Medical Physiology, Texas A&M University College of Medicine, Bryan, TX, USA. [3]Center for Human Nutrition, University of Texas Southwestern Medical Center, Dallas, TX, US. [4]Renal, Electrolyte, and Hypertension Division, Department of Medicine, University of Pennsylvania Perelman School of Medicine, Philadelphia, PA 19104, USA. [5]Department of Nephrology and Medical Intensive Care, Charité, Universitätsmedizin Berlin, 10117 Berlin, Germany. [6] Berlin Institute of Health at Charité, Universitätsmedizin Berlin, BIH Biomedical Innovation Academy, BIH Charité Clinician Scientist Program, 10117 Berlin, Germany. [7]Department of Anesthesiology, Critical Care and Pain Medicine, UT Health Science Center at Houston, Houston, TX, USA. [8]Center for Hypothalamic Research, University of Texas Southwestern Medical Center, Dallas, TX, USA. [9]Solutions COE, Analytical & Measuring Instruments Division, Shimadzu Corporation, Kyoto, Japan. [10]Liver Tumor Translational Research Program, Simmons Comprehensive Cancer Center, Division of Digestive and Liver Diseases, Department of Internal Medicine, University of Texas Southwestern Medical Center, 5323 Harry Hines Blvd, Dallas, TX 75390, USA. [11]Departments of Cell Biology and Internal Medicine, University of Texas Southwestern Medical Center, Dallas, TX, USA. [12]Charles and Jane Pak Center for Mineral Metabolism and Clinical Research, University of Texas Southwestern Medical Center, Dallas, TX, USA. [13]Department of Internal Medicine, University of Texas Southwestern Medical Center, Dallas, TX, USA. [14]Department of Physiology, University of Texas Southwestern Medical Center, Dallas, TX, USA. ✉e-mail: Philipp.Scherer@UTSouthwestern.edu

