## [Peer Review File · Nature Communications]

Endogenous renal adiponectin drives gluconeogenesis through enhancing pyruvate and fatty acid utilizationEditorial Note: Parts of this Peer Review File have been redacted as indicated to remove third-party material where no permission to publish could be obtained.

REVIEWER COMMENTS

Reviewer #1 (Remarks to the Author):

The authors described in this manuscript that kidney-specific adiponectin KO mice exhibited an impaired gluconeogenesis.

This is a well-written paper containing interesting results which merit publication. For the benefit of the reader, however, there are many critical and important points to be addressed, as listed below.

1) The authors showed that adiponectin activated mitochondrial function and gluconeogenesis. They should examine whether the adiponectin-induced increase in renal gluconeogenesis would be mediated by AdipoR1 or AdipoR2.

2) Related to point 1), the authors should discuss the present experimental results based on previous reports from the first report on the association of AdipoR1, AdipoR2 and T-cadherin with adiponectin.

3) The KSPAPN mice shown in this MS, in which excessive adiponectin is secreted from the epithelia of the proximal tubules to the collecting ducts, appeared to exhibit a highly non-physiological phenotype. In fact, despite a 4-fold increase in blood adiponectin levels, systemic glucose intolerance worsened, contradicting previous reports. The authors should explain this discrepancy.

4) In Fig. 1a, the authors compared the expression of adiponectin in the kidney with that of the heart and other organs, but it should also be compared with adipose tissue so that the reader is not misled.

Reviewer #2 (Remarks to the Author):

Onodera et al. show a role of endogenous renal adiponectin in regulating renal gluconeogenesis through pyruvate and fatty acid oxidation. This work highlights the importance of kidney metabolism, especially gluconeogenesis, in regulating systemic metabolism. The authors have provided sufficient evidence to support their conclusion. However, I have some remaining remarks that need to be addressed:

1. In line 556, they described that q-pcr was done on the samples obtained from both cortex and inner medulla. In they described that “Consistent with the qPCR data, renal adiponectin expression was detected in the components that reside in the medulla, including in the proximal straight tubule...”. In fact, there are no proximal tubules in inner medulla, but in outer medulla and cortex.
2. In line 601, It’s not clear what “this area” refer to. And from figure S3E-J, it’s difficult to say which area shows the highest degree of the lipid uptake-related gene expression.
3. From single cell RNA-seq data of adult kidney, the gluconeogenic genes are predominantly expressed in proximal tubules, while adiponectin is expressed in proximal straight tubule, but not proximal convoluted tubule. Is the adiponectin regulated gluconeogenesis specific for proximal straight tubule? Or work for both through release of adiponectin? What’s the effect of adipocyte released adiponectin on proximal tubule gluconeogenesis?
4. In line 606, In the sample of mouse at birth (P0), “We detected expression of renal adiponectin in the PST and IC, but not in proximal convoluted cells (PCT) (Fig. 1L).” However, it showed expression of adiponection in PCT-P0 in Fig. 1L.
5. Line 643-651, the description of KSPAPN model should be together with figure 2A?
6. In figure 3, overexpression of adiponetin increased serum adiponectin, which may also affect the results of OGTT and ITT through other tissues, such as adipocyte and muscle. So is the cold fasting induced adiponectin expression in kidney have a similar level of adiponetin expression as the KSPAPN model? How good is this KSPAPN model representing a physiology ? Will overexpression of adiponetin in other cells like adipocytes have similar effects on kidney metabolism through serum adiponetin level as well?
7. In figure 4H-I, there are no significant differences on the TCA intermediates between control and KSPAPN, while Akg is significantly different in liver. In line 725, It reads “A similar

trend for increased lactate and TCA cycle metabolites were observed in KSPAPN kidney (Fig. 4H) but not liver tissue (Fig. 4I).” This conclusion may mislead the readers without carefully reading the figures.

8. In line 730-736, “Blood lactate ^{13}C enrichment in KSPAPN mice correlated strongly with blood.... These data indicate that differences in the plasma profile of KSPAPN mice originates from kidney” It is known that kidney take up serum lactate and citrate as fuel. This observation may also be due to metabolite uptake. This conclusion is therefore not accurate.

9. The authors show both pyruvate and fatty acid oxidation related gluconeogenesis in adiponectin overexpression model. How are pyruvate and fatty acid related? Will fatty acid oxidation results in more pyruvate production?

10. It’s known that kidney tubules rely on fatty acid metabolism mainly. How to explain the results of seahorse experiment that inhibition of CPT1 had no effect on fatty acid oxidation in control samples?

11. How does the pyruvate metabolism change in KSPAKO kidneys? Such as the pyruvate tolerance test. The authors test several metabolites on this model, but missed this very important one.

12. How does the fatty acid oxidation change in KSPAKO kidneys? This is also very important message to support their main conclusion.

13. In line 872, “The most striking changes of lipids were observed in the transition area, which is consistent with the fact that renal adiponectin is highly expressed in the distal tubules which primarily constitute the transition area.” It should not be mainly distal tubules in this area. Please confirm this with IF staining.

14. What’s the mass resolution of your MALDI-TOF measurements? In general, it’s not sufficient to identify the lipid species using MALDI-TOF only. Additional techniques are needed for lipid identifications, such as MALDI-FTICR or LC-MS/MS.

Reviewer #3 (Remarks to the Author):

The present manuscript examines the impact of kidney-specific overexpression or knockout of adiponectin. The kidney-specific overexpression is quite substantial and changes circulating adiponectin levels markedly. There is a lot of scholarly data provided on this

model. Most effect sizes are modest and without controls with analogous changes in systemic adiponectin without impacting the kidney, it was hard for me to figure out whether they are mediated by local or systemic effects. In any event, these data do not directly support the title which relates to endogenous renal adiponectin.

The endogenous renal adiponectin data is provided Fig 6/7, again with modest effect sizes except for the data on fibrotic and inflammatory gene expression, which is striking. There also interesting data on changes in spatial lipids in the kidney, which are state-of-the-art although accuracy of metabolite identification is an open question.

Specific concerns:

1. The differences in circulating glucose response to different substrates (e.g. more sensitive to glucose itself and 3HB, equally sensitive to glycerol, less sensitive to glutamine) for renal adiponectin overexpression are intriguing and impressively comprehensive analyses for impact of different substrates on glucose across 2 genotypes. On the flipside, the decision to focus on pyruvate is not clear, when 3HB and glutamine seem to give the strongest effects (albeit in opposite directions). Can the authors explain?
2. Conclusions regarding the absolute fractional contribution of one substrate to another are, from my perspective, technically correct but not really capturing the relevant metabolic phenomena. I understand the calculation, and it confounds concentrations and fluxes. If one substrate comes more or less solely from another (e.g. lactate from pyruvate), then any time lactate blood concentration goes up the “absolute blood amount of lactate from pyruvate” will also be up-- this is spelled out in the definition but probably not how most people are going to interpret this? The more natural interpretation is what’s happening to the actual flux from circulating pyruvate to circulating lactate. I’m not going to insist on how these distinguished authors handle this issue in the current paper, but at a minimum they should be very clear about the limitations and potential misinterpretations of the current approach, and how it differs from approach that actually calculates fluxes between circulating metabolites.
3. Statistics: there seem to be 3 problems: (i) no correction for multiple statistical comparisons (e.g. if you test 4 timepoints x 5 substrate tolerances, by chance alone you would expect 1 timepoint to show significant difference, which is perilously close to what seen in Fig 6). (ii) no individual data points shown for some key figures, e.g. tolerance tests.

(iii) when individual data is shown, sometimes it is normally distributed, in which case statistics are appropriate, and sometimes it is very far from normally distributed, e.g. 1I/J, in which case a nonparametric statistic or transformation to render the data roughly normal are needed. In such cases, often the non-parametric analysis (or parametric analysis after, e.g. log transformation) will actually show a stronger effect, because the lack of normality leads to overestimation of the relevant error, but we need to see.

4. The imaging mass spectrometry is intriguing, but some of the measured metabolites are hard to understand, e.g. what is HexCer 37:6; O5? I understand everything until the O5, but where the oxygen supposed to be? In general, there a lot of lipids reported with oxygens. Are these extra oxygens beyond those typically found in the relevant lipid? If so, what kind of structures are being proposed and how can these be validated? Also, the authors should be very clear that the observed lipids likely include fragments, e.g. DG signals may reflect TG degradation during the laser ablation.

Minor: abstract line 58, it says lower utilization and accumulation of lipids. I think the authors mean lower utilization and greater accumulation?

Line 108: literature data on energy sources in different parts of the kidney are not all in alignment. Recent in vivo data from imaging mass spectrometry with isotope tracing show relatively little fat use in the cortex, more in the medulla, with the cortex heavily reliant on organic acids.

Methods: please be more clear on how the kidney was prepared for the Seahorse analyses

Line 897: functional data are lacking to show that altered lipid levels mediate the lower mitochondrial function and fibrotic phenotype in the KSPAKO kidneys.

Line 957: agreed that parts of the kidney probably produce while others probably consume glucose, although many papers show that under pretty typical fasted conditions the kidney is net gluconeogenic

Replies to Reviewers' comments

We thank all of the reviewers for their time and careful assessment of our manuscript. We were pleased to see many positive remarks about the data presented, and truly appreciate the critical comments and constructive suggestions, which have been extraordinarily helpful in preparing a revised version of this manuscript. We have performed a number of new experiments to address the concerns. Major changes were highlighted in the revised manuscript.

REVIEWER COMMENTS

Reviewer #1 (Remarks to the Author):

The authors described in this manuscript that kidney-specific adiponectin KO mice exhibited an impaired gluconeogenesis.

This is a well-written paper containing interesting results which merit publication. For the benefit of the reader, however, there are many critical and important points to be addressed, as listed below.

- **Major1.** The authors showed that adiponectin activated mitochondrial function and gluconeogenesis. They should examine whether the adiponectin-induced increase in renal gluconeogenesis would be mediated by AdipoR1 or AdipoR2.

Response: Thank you for the important remarks. To check whether the impact of circulating adiponectin on the renal gluconeogenesis phenotype, we assessed the renal gluconeogenic phenotype by glutamine tolerance test and renal gluconeogenic enzyme mRNA expression in AdipoR2 KO mice. We also added the data on adiponectin receptors and T-cadherin in the kidney cells (**Fig. S5A**). As shown in **Fig. S5B and C**, the global deficiency of AdipoR2 showed higher glucose levels during a glutamine tolerance test and had a minimal impact on the gluconeogenic phenotype in the kidney. Global AdipoR1 deficiency exhibits higher body weight and glucose intolerance¹. Therefore, adiponectin signaling through AdipoR1 should not be relevant for the gluconeogenic phenotype.

- **Major2.** Related to point 1), the authors should discuss the present experimental results based on previous reports from the first report on the association of AdipoR1, AdipoR2 and T-cadherin with adiponectin.

Response: According to the reviewer's suggestion, we added the expression data in **Fig. S5A** and inserted a discussion about AdipoR1, AdipoR2 and T-cadherin starting with **line 1372**.

The inserted discussion is as follows:

Adiponectin exerts its beneficial effects through its receptors AdipoR1, AdipoR2 and T-cadherin². AdipoR1 and 2 were identified as cell surface receptors and utilize ceramides and AMPK as the downstream signaling molecules^{3,4}. T-cadherin is also a molecule that has affinity for hexameric and multimeric adiponectin, but lacks a cytoplasmic signaling domain⁵. AdipoR1 and AdipoR2 are widely distributed in the kidney and T-cadherin is mainly expressed in kidney endothelial cells. Clinically, both adiponectin and adiponectin receptors are up-regulated in end-stage kidney disease to compensate the functional deterioration of the kidney⁶. The adiponectin receptor agonist adiponon ameliorates diabetic nephropathy⁷, suggesting that signal transduction through AdipoRs prevents kidney injury. A global KO of T-cadherin shows

worsened acute kidney injury, consistent with the adiponectin KO mouse phenotype⁸. On the other hand, we confirmed that AdipoR2 is not relevant to the renal gluconeogenic pathway because the glucose levels in glutamine tolerance tests were even higher and gluconeogenic genes were not altered in the AdipoR2 KO mice. Given that the AdipoRs are mainly localized in the plasma membrane⁴, these data suggest that the signals from circulating adiponectin are vital for the prevention of renal injury but not for the renal gluconeogenesis pathway.

• **Major3.** The KSPAPN mice shown in this MS, in which excessive adiponectin is secreted from the epithelia of the proximal tubules to the collecting ducts, appeared to exhibit a highly non-physiological phenotype. In fact, despite a 4-fold increase in blood adiponectin levels, systemic glucose intolerance worsened, contradicting previous reports. The authors should explain this discrepancy.

Response: An excellent point. Previously, the overexpression of adiponectin was done in liver. As for adipose tissue, not a full length but a slightly truncated version of adiponectin lacking the collagen domain was successfully overexpressed. Adiponectin overexpression in the liver results in reduced body weight and improved glucose metabolism⁹. On the other hand, adiponectin overexpression in adipose tissue manifests a higher body weight phenotype¹⁰ although glucose metabolism was improved in both models of adiponectin overexpression. Therefore, we propose that the *location* of adiponectin overexpression substantially affects the systemic phenotype, including body weight and glucose metabolism. The basic function of adiponectin is to enhance fatty acid oxidation, but the consequences of enhanced fatty acid oxidation vary depending on the organ. The kidney enhances gluconeogenesis, but this clearly does not occur in the liver.

• **Major4.** In Fig. 1a, the authors compared the expression of adiponectin in the kidney with that of the heart and other organs, but it should also be compared with adipose tissue so that the reader is not misled.

Response: Thank you for the important remark. We demonstrate the functional involvement of kidney adiponectin in glucose metabolism in this manuscript. Nevertheless, it is true that adiponectin expression is much more prominently expressed in adipose tissue and serves as an adipocyte marker. As the reviewer suggested, we included the adiponectin expression data of epididymal adipose tissue in **Fig. 1A** that shows a comparison of adiponectin expressions across the various organs. We have to bear in mind that adiponectin is only expressed in a subset of kidney cells as opposed to the high level expression across all adipocytes.

Reviewer #2 (Remarks to the Author):

Onodera et al. show a role of endogenous renal adiponectin in regulating renal gluconeogenesis through pyruvate and fatty acid oxidation. This work highlights the importance of kidney metabolism, especially gluconeogenesis, in regulating systemic metabolism. The authors have provided sufficient evidence to support their conclusion. However, I have some remaining remarks that need to be addressed:

• **Major 1.** In line 556, they described that q-pcr was done on the samples obtained from both cortex and inner medulla. In they described that “Consistent with the qPCR data, renal adiponectin expression was detected in the components that reside in the medulla, including in

the proximal straight tubule...". In fact, there are no proximal tubules in inner medulla, but in outer medulla and cortex.

Response: Thank you for the comments. We noticed that we did not properly describe our qPCR efforts. For the qPCR of cortex and medulla, we separated the kidney into cortex and medulla, including outer and inner medulla. Taking into consideration both the histology that we will mention in **Major 13**, qPCR and single-cell sequencing results, we concluded that expression of adiponectin in the Loop of Henle, connecting tubule and collecting duct in outer and inner medulla, contributed to the higher expression of adiponectin in the medulla over the cortex. As the reviewer suggested, we corrected the sentences from **line 668** into "Consistent with the qPCR data, renal adiponectin expression was detected in the components that reside in the outer medulla and the inner medulla,". Also, we corrected the sentences from line 616 to "Comparing the adiponectin expression between the cortex and the medulla reveals that adiponectin mRNA is higher in the medulla rather than in the cortex (**Fig. 1E**)."

• **Major 2.** In line 601, It's not clear what "this area" refer to. And from figure S3E-J, it's difficult to say which area shows the highest degree of the lipid uptake-related gene expression.

Response: According to the reviewer's suggestion, we specify the term "this area" clearly. In **line 672**, we corrected as follows. "Particularly, CNT and collecting duct cells including PC, IC and transitional cells shows the highest degree of the lipid uptake-related gene expression (**Fig. S3E-J**)."

• **Major 3.** From single cell RNA-seq data of adult kidney, the gluconeogenic genes are predominantly expressed in proximal tubules, while adiponectin is expressed in proximal straight tubule, but not proximal convoluted tubule. Is the adiponectin regulated gluconeogenesis specific for proximal straight tubule? Or work for both through release of adiponectin? What's the effect of adipocyte released adiponectin on proximal tubule gluconeogenesis?

Response: Since the gluconeogenic enzyme expression in the kidney is limited to the proximal tubules, including proximal convoluted tubules and straight tubules, we assume there are two possibilities for the enhancement of gluconeogenesis by renal adiponectin. First, as the Reviewer points out, the expression of adiponectin and phosphoenolpyruvate carboxykinase 1 (Pck1) overlap in the proximal straight tubule. Adiponectin directly increases the expression of Pck1 and enhances gluconeogenic activity in the proximal straight tubules. Another possibility is the existence of a transport system for substrates, such as lactate, from the distal tubule or collecting duct to the proximal tubule. This possibility is supported by the fact that kidney adiponectin is more highly expressed between the ascending limb of the Loop of Henle to the collecting duct rather than the proximal tubules. In parallel, Ksp-driven overexpression of adiponectin induces higher adiponectin levels from the ascending limb of the Loop of Henle to the collecting duct. Previous reports pointed out that the production and consumption of substrates can occur simultaneously in the kidney ¹¹. Lactate is more highly enriched in the medulla, while lactate is mainly consumed in the cortex ^{12,13}, suggesting a lactate transport system from the medulla to the cortex. These lactate transport systems by the monocarboxylate transporter1 and 4 (Mct1 and 4) are active in the kidney ¹⁴. Then, lactate produced in a paracrine fashion can be utilized for gluconeogenesis in the proximal tubules, while lactate production is localized in distal tubule and the collecting duct. Given that adiponectin is involved in fatty acid oxidation and lactate production, we assume the 2nd possibility is more likely to apply.

AdipoR1, AdipoR2 and T-cadherin are mainly expressed on the cell surface and mediate the function of circulating adiponectin. To test the possibility of the involvement of adiponectin receptors in renal gluconeogenesis, we checked the kidney gluconeogenic phenotype in AdipoR2 KO mice by glutamine tolerance tests and qPCR. We believe that AdipoR2 has no contribution to the renal gluconeogenesis (**Fig. S5B and C**).

• **Major 4.** In line 606, In the sample of mouse at birth (P0), “We detected expression of renal adiponectin in the PST and IC, but not in proximal convoluted cells (PCT) (Fig. 1L).” However, it showed expression of adiponectin in PCT-P0 in Fig. 1L.

Response: Thank you so much for pointing out this contradictory statement. According to the reviewer’s suggestion, we corrected the sentence as follows in **line678**; “We detected expression of renal adiponectin in the proximal straight tubule (PST), intercalated cells (IC) and proximal convoluted cells (PCT) (**Fig. 1L**).”.

• **Major 5.** Line 643-651, the description of KSPAPN model should be together with figure 2A?

Response: As suggested by the reviewer, we moved the description of KSPAPN mice in Fig.3 to the result section of Fig. 2A. The updated portion of the result section of Fig. 2 is as follows, starting with **line696** :

“To determine the *in vivo* functional role of renal adiponectin, inducible Ksp^{rt}TA/TRE-Adiponectin mice (KSPAPN) were used to examine the consequences of inducible adiponectin overexpression specifically in kidney epithelial cells. Expression is induced upon administration of doxycycline. Ksp^{rt}TA drives the expression of genes in the epithelia from the proximal tubule to the collecting duct in both the adult and in the developing kidney ¹⁵ (**Fig. 2A**). As a first step, we performed RNAseq analysis on KSPAPN in which we overexpress adiponectin in a doxycycline-inducible manner.”

• **Major 6.** In figure 3, overexpression of adiponectin increased serum adiponectin, which may also affect the results of OGTT and ITT through other tissues, such as adipocyte and muscle. So is the cold fasting induced adiponectin expression in kidney have a similar level of adiponectin expression as the KSPAPN model? How good is this KSPAPN model representing a physiology? Will overexpression of adiponectin in other cells like adipocytes have similar effects on kidney metabolism through serum adiponectin level as well?

Response: According to the reviewer’s comment, we measured serum adiponectin level after fasting and cold exposure (**Fig. S1G**). The concentration of serum adiponectin is significantly reduced after fasting and cold exposure. Although we could detect a strong up-regulation of kidney adiponectin expression (**Fig. 1G**), this up-regulation can be a compensatory mechanism for decreased serum adiponectin levels and is also beneficial to enhance renal gluconeogenesis. We described these results from **line625**.

The phenotypes of adiponectin overexpression vary depending on the sites where adiponectin is overexpressed. As we discussed in **Reviewer1 Major 3**, adiponectin overexpression in the liver results in reduced body weight and improved glucose metabolism ⁹. On the other hand, adiponectin overexpression in adipose tissue manifests a higher body weight phenotype ¹⁰ although glucose metabolism was alleviated in both models of adiponectin overexpression. Kidney-specific adiponectin overexpression shows a higher glucose phenotype without a body

weight change. It is true that increased levels of serum adiponectin will stimulate adiponectin receptors on peripheral organs and exert beneficial effects. Nevertheless, we assume that the local effects of adiponectin overexpression strongly determine the systemic glucose metabolism.

• **Major 7.** In figure 4H-I, there are no significant differences on the TCA intermediates between control and KSPAPN, while Akg is significantly different in liver. In line 725, It reads “A similar trend for increased lactate and TCA cycle metabolites were observed in KSPAPN kidney (Fig. 4H) but not liver tissue (Fig. 4I).” This conclusion may mislead the readers without carefully reading the figures.

Response: We thank the reviewer for pointing out this inaccuracy. We corrected the text, so it more accurately describes these panels. Now it reads: “Plasma TCA cycle intermediates were elevated in KSPAPN mice (**Fig. 4H**), though these metabolites are expected to be in low concentration compared to tissue concentrations. TCA cycle intermediate concentrations tended to be higher in the kidney of KSPAPN mice, but the differences did not reach significance (**Fig. 4H**), and only Akg was increased in the liver of KSPAPN mice (**Fig. 4H**).”

• **Major 8.** In line 730-736, “Blood lactate 13C enrichment in KSPAPN mice correlated strongly with blood.... These data indicate that differences in the plasma profile of KSPAPN mice originates from kidney” It is known that kidney take up serum lactate and citrate as fuel. This observation may also be due to metabolite uptake. This conclusion is therefore not accurate.

Response: Thank you for this insightful comment. We agree that distinguishing the directionality of this process is difficult in this experiment. Therefore, we removed most of the dialogue regarding lactate production by the kidney. Although we retain the correlations in the supplemental data, we now state that “*These data suggest plasma lactate originates from or contributes to kidney metabolites in KSPAPN mice.*” Instead, we focus more deeply on the fractional contribution of pyruvate carbons to TCA cycle intermediates and glucose (See response to **R3 point 3**).

However, based on the reviewer’s comment, we attempted a more rigorous modeling of substrate flux from the isotopologue data. We show it here (**Figure R2.9.1**) for the reviewer but ultimately decided against reporting it due to inherent limitations of modeling a bolus injection, the involvement of multiple compartments and relatively poor SSR for the regression analysis.

[FIGURE REDACTED]

Figure R2.9.1

To investigate the source of blood lactate, we undertook metabolic flux analysis (MFA) approach using INCA 2.1 software (**Figure R2.9.1**). Briefly, we created a series of tissue specific and multicompartement models and used all MID data from all metabolites in a single data regression.

We found that during the pyruvate bolus, the data regressed better to a model where most pyruvate is taken up by the liver and converted to glucose. Conversely, the model predicted that pyruvate metabolized in the kidney was partially converted to lactate and exported to the blood compartment. We note that this analysis resulted in a range of solutions depending on how pyruvate was divided between liver and kidney which could not be determined from the data. Nonetheless, most solutions used the kidney compartment as a source of blood lactate (**Figure R2.9.1**). A more detailed description is provided in the response to point 2 by reviewer #3. It is possible that the unexpected release of lactate is due to the nature of the pyruvate bolus, but in light of the reviewer's insight, the complexity of the experiment, and the relatively poor SSR of the multicompartement regression analysis (see below, **Figure R3.3.2**), we excluded the modeling data and most dialogue about kidney lactate production.

• **Major 9.** The authors show both pyruvate and fatty acid oxidation related gluconeogenesis in adiponectin overexpression model. How are pyruvate and fatty acid related? Will fatty acid oxidation results in more pyruvate production?

Response: Thank you for this question. The connection between increased fat oxidation and elevated gluconeogenesis has long been recognized in liver¹⁶. There are several factors that are important but in brief, fat oxidation promotes GNG by increasing allosteric mediators, such as acetyl-CoA which allosterically activates pyruvate carboxylase and the conversion of pyruvate to oxaloacetate which is used by PEPCK to produce PEP for gluconeogenesis. In liver, fat oxidation is thought to increase cytosolic citrate which allosterically suppresses PFK. More generally, increased substrate oxidation supplies more ATP, which is necessary for gluconeogenesis.

• **Major 10.** It's known that kidney tubules rely on fatty acid metabolism mainly. How to explain the results of seahorse experiment that inhibition of CPT1 had no effect on fatty acid oxidation in control samples?

Response: For the Seahorse experiments, we harvested kidney tissue, cut it into half longitudinally and isolated a 2mm biopsy punch. Under our experimental conditions, the supplementation of palmitate and carnitine to control mice had a minimal impact on the OCR measured by the Seahorse assay. The up-regulation of OCR by palmitate becomes evident only after the overexpression of adiponectin in the kidney. It is absolutely correct that the kidney utilizes fatty acids as the main source of energy *in vivo*. However, these data suggest that fatty acids are not incorporated actively under *ex vivo* conditions without overexpression of adiponectin. This is to us the only viable explanation why the addition of the Cpt1 inhibitor has no effect on the control samples.

• **Major 11.** How does the pyruvate metabolism change in KSPAKO kidneys? Such as the pyruvate tolerance test. The authors test several metabolites on this model, but missed this very important one.

Response: Under normal physiological conditions, the lactate to pyruvate ratio is approximately 10:1. So, to see the impact of pyruvate and lactate on gluconeogenesis, we originally performed

lactate/pyruvate tolerance tests to recapitulate the physiological conditions rather than pyruvate tolerance tests alone. Many papers utilize this approach with lactate/pyruvate tolerance tests instead of pyruvate tolerance test alone¹⁷. In our paper, in order to clarify the conversion from pyruvate to lactate, we did take advantage of a pyruvate tolerance test for the flux experiments. As the reviewer suggests, we now also performed a pyruvate tolerance test using kidney adiponectin KO mice. To be consistent with the tolerance tests performed with kidney adiponectin overexpression mice, we replaced the lactate pyruvate tolerance test with pyruvate tolerance test in **Fig.6M**.

• **Major 12.** How does the fatty acid oxidation change in KSPAKO kidneys? This is also very important message to support their main conclusion.

Response: Thank you for bring up this interesting suggestion. As the reviewer requested, we now include the fatty acid incorporation data in KSPAPN kidney (**Fig. S7A, B and C**) and fatty acid oxidation data in KSPAKO kidney (**Fig. S7D, E and F**). Overall, the utilization of fatty acids by the kidney is enhanced by renal adiponectin overexpression and down-regulated in the renal adiponectin KO. However, the data shows that the effects of overexpression and KO is not perfectly symmetrical. Specifically, overexpression of adiponectin in the kidney enhances fatty acid oxidation rather than incorporation of fatty acid to the cells. On the other hand, renal adiponectin deficiency reduces the uptake of fatty acid into the cells while the fatty acid oxidation was not altered.

• **Major 13.** In line 872, “The most striking changes of lipids were observed in the transition area, which is consistent with the fact that renal adiponectin is highly expressed in the distal tubules which primarily constitute the transition area.” It should not be mainly distal tubules in this area. Please confirm this with IF staining.

Response: Thank you for this important remark. We stained the WT kidney tissues with LTL (proximal tubule marker), Umod (Ascending limb of Loop of Henle, straight and convoluted distal tubule marker) and Calbindin1 (distal convoluted tubule, connecting tubule and collecting duct marker) antibodies (**Fig. S9**). The transition area is mainly composed of LTL⁺ cell and Umod⁺ cells. There is also a calbindin 1⁺ cell component in the transition area, but the contribution is less compared to the LTL⁺ cell and Umod⁺ cells. Overall, the transition area is composed of mainly proximal tubules, the ascending Limb of loop of Henle and distal tubules, while the renal lipid species are distributed in a more stratified fashion. Based on this data, we corrected the sentences from **line1141** as follows:

To identify the tubular components of the transition area in the kidney, we performed immunofluorescence with 3 markers, including lotus tetragonolobus (LTL) (proximal tubule marker), uromodulin (Umod) (ascending limb of the Loop of Henle, straight and convoluted distal tubule marker) and calbindin1 (Calb1) (distal convoluted tubule, connecting tubule and collecting duct marker)¹⁸. The immunofluorescence showed that the transition area is mainly composed of LTL⁺ cells and Umod⁺ cells. This data indicate that transition area is mainly comprised of the proximal tubule, the ascending limb of the Loop of Henle and distal tubules, which partially overlaps with the distribution of endogenous adiponectin from proximal straight tubules to convoluted distal tubules (**Fig. S9**).

• **Major14.** What’s the mass resolution of your MALDI-TOF measurements? In general, it’s not sufficient to identify the lipid species using MALDI-TOF only. Additional techniques are needed for lipid identifications, such as MALDI-FTICR or LC-MS/MS.

Response: Thank you for this important and well-informed remark. Our instrument has a resolution of 30,000 FWHM and a mass accuracy of <1 ppm at m/z 622.5662. As the reviewer mentioned, MALDI-FTICR (Fourier-transform ion cyclotron resonance) mass spectrometer is a massive piece of instrumentation, extremely expensive and very hard to maintain. Only a few labs maintain one of these instruments and it has a huge 7.0 Tesla magnet in the back. It is a fact that this instrumentation provides unbeatable mass resolution. We are cognizant of the fact that there is a technical limitation for the assignment of the lipid species and LC-MS/MS experiments are necessary for unequivocal lipid assignment and fatty acid composition. However, this is not the scope of our work. Running LC-MS/MS experiments and validating every lipid reported by mass spec imaging is rather laborious and would require years of labor-intensive work. Therefore, to capture the whole landscape of lipid alterations by adiponectin overexpression and deficiency, we provide the data as a heatmap by adding up the same group of lipid species after removing the species that has overlapping annotations (**Fig. 7I**). Also, we provide heatmap data that only includes the lipid species whose existence is validated by LC-MS/MS in the kidney (**Fig. S10**).

Reviewer #3 (Remarks to the Author):

The present manuscript examines the impact of kidney-specific overexpression or knockout of adiponectin. The kidney-specific overexpression is quite substantial and changes circulating adiponectin levels markedly. There is a lot of scholarly data provided on this model. Most effect sizes are modest and without controls with analogous changes in systemic adiponectin without impacting the kidney, it was hard for me to figure out whether they are mediated by local or systemic effects. In any event, these data do not directly support the title which relates to endogenous renal adiponectin.

The endogenous renal adiponectin data is provided Fig 6/7, again with modest effect sizes except for the data on fibrotic and inflammatory gene expression, which is striking. There also interesting data on changes in spatial lipids in the kidney, which are state-of-the-art although accuracy of metabolite identification is an open question.

Specific concerns:

- **Major 1.** The differences in circulating glucose response to different substrates (e.g. more sensitive to glucose itself and 3HB, equally sensitive to glycerol, less sensitive to glutamine) for renal adiponectin overexpression are intriguing and impressively comprehensive analyses for impact of different substrates on glucose across 2 genotypes. On the flipside, the decision to focus on pyruvate is not clear, when 3HB and glutamine seem to give the strongest effects (albeit in opposite directions). Can the authors explain?

Response: Thank you raising this important point. We also recognize the up-regulation of blood glucose by 3HB administration in kidney adiponectin overexpression mice as an interesting phenomenon. The reduced impact of glutamine in the adiponectin overexpression mice is important as well. It is reported that 3-HB facilitates the renal gluconeogenesis¹⁹. The result that 3HB administration increases blood glucose levels implicates enhanced renal gluconeogenesis, but 3HB itself is not converted to glucose. Comparisons of the tolerance test phenotypes in the adiponectin overexpression and KO mice indicates that the pyruvate tolerance test and alanine tolerance test exhibit a symmetrical phenotype between the two models. Based on these results, we assume that adiponectin impacts a step in the gluconeogenic pathway of alanine to oxaloacetate through pyruvate. While glutamine enters the TCA cycle from alpha-ketoglutarate,

this pathway does not involve adiponectin. Although glutamine is well known as a substrate specific for renal gluconeogenesis, pyruvate is utilized unspecifically but preferentially utilized by renal gluconeogenesis rather than glutamine. These are the reasons why we focused primarily on pyruvate mediated gluconeogenesis.

- **Major 2.** Conclusions regarding the absolute fractional contribution of one substrate to another are, from my perspective, technically correct but not really capturing the relevant metabolic phenomena. I understand the calculation, and it confounds concentrations and fluxes. If one substrate comes more or less solely from another (e.g. lactate from pyruvate), then any time lactate blood concentration goes up the “absolute blood amount of lactate from pyruvate” will also be up-- this is spelled out in the definition but probably not how most people are going to interpret this? The more natural interpretation is what’s happening to the actual flux from circulating pyruvate to circulating lactate. I’m not going to insist on how these distinguished authors handle this issue in the current paper, but at a minimum they should be very clear about the limitations and potential misinterpretations of the current approach, and how it differs from approach that actually calculates fluxes between circulating metabolites.

We thank the reviewer for this insightful perspective, upon which we agree. As the reviewer points out, metabolite enrichments, isotopologues, or isotopomers are frequently reported under the umbrella of “flux” but, on their own, can be challenging to interpret. For example, the data obtained from the [U-¹³C₃]pyruvate bolus presents multiple challenges. First, observing the incorporation of pyruvate carbons into downstream pathways is akin to a substrate competition experiment. We can only infer the rate of pathways that incorporate enrichment RELATIVE to pathways that dilute the enrichment. The precursor-product analysis we use here is helpful in evaluating sources of carbon atoms (or fractional contributions) from one metabolite to another. However, these are not rates and are not easily extrapolated to rates because we do not know the rate at which the tracer is delivered to the system. In the results section, we now explicitly state:

“To examine the fate of pyruvate during the gavage, we calculated the fractional contribution (FC) of pyruvate to various downstream metabolites. The total FC of pyruvate to downstream metabolites was calculated as the ratio of atom percent enrichment (APE) of a metabolite to the APE of circulating pyruvate. The FC specific to exogenous pyruvate was calculated as the ratio of a metabolite’s APE to exogenous pyruvate APE (i.e., 40%). The difference between these values represents the FC from sources that do not enter the pyruvate pool. Although this calculation does not report metabolic rates, it provides insight into the fate of pyruvate carbons following the tracer injection.”

Second, there is an important distinction between pyruvate conversion to a metabolite and the incorporation of pyruvate carbons into a metabolite. The former scenario reduces to a simple competition between pyruvate and all other substrates. In contrast, the latter scenario is more challenging to interpret and, unfortunately, more common than the former scenario. For example, glucose produced from [U-¹³C₃]pyruvate will always be diluted with non-pyruvate carbon, even if pyruvate is the only substrate for net glucose synthesis. Of course, this occurs because pyruvate is initially carboxylated to OAA, a TCA cycle metabolite in rapid turnover and dilution by acetyl-CoA oxidation (via citrate synthase) before its ultimate conversion to glucose. A rigorous metabolic flux analysis would result in the expected conclusion that 100% of glucose was derived from pyruvate. However, calculating the fractional contribution of pyruvate carbons to glucose would indicate that a significant portion of glucose carbons were derived from non-pyruvate sources, when in fact, those unlabeled carbons were derived from metabolites that

cannot have a net contribution to glucose synthesis (i.e., bicarbonate and acetyl-CoA). Thus, in this kind of experiment, it is difficult to say whether glucose labeling is altered by the contribution of pyruvate relative to another precursor (e.g., glutamine or glycerol) or by different relative rates of pyruvate carboxylation and citrate synthase flux. It is entirely plausible that higher rates of pyruvate conversion to glucose can result in lower glucose enrichment if citrate synthase flux is disproportionately increased. We added the following text to highlight this point:

“The FC of exogenous pyruvate to some TCA cycle intermediates was significantly higher in KSPAPN blood (Fig. S4A) and liver (e.g., malate) (Fig. S4B), but surprisingly, not in the kidney (Fig. S4C). It is important to note that the specific contribution of pyruvate to kidney TCA cycle intermediates was lower than in the liver regardless of genotype (Fig. S4B versus S4C). Thus, non-pyruvate anaplerotic substrates (e.g., glutamine) or higher citrate synthesis from unlabeled acetyl-CoA (e.g., via fat oxidation) relative to anaplerosis must dilute kidney TCA cycle intermediates compared to the liver. Unfortunately, the method will not detect changes in pyruvate anaplerosis in KSPAPN kidney if these pathways are systematically altered, for example, by proportional increases in pyruvate anaplerosis and fat oxidation, which we address below.”

Thus, the best we can do in the present case is to quantify the fraction of carbons from endogenous pyruvate, exogenous pyruvate, and non-pyruvate sources (which includes dilution in the TCA cycle) and ask whether the glucose excursion can be differentially attributed to these sources and whether the implicated source better resembles TCA cycle intermediates in the liver or kidney. Based on the reviewer’s suggestion, we changed how the data is reported. While we kept some of the absolute (FC x concentration) panels, we changed their description. Rather than referring to this data as “gluconeogenesis,” which could be misinterpreted as a rate, we now title the data as blood glucose concentration, broken down by fractional contributions of carbon sources. In short, showing that the rise in blood glucose can be attributed to exogenous pyruvate and unlabeled sources has limitations, but we believe it is more valuable than reporting only isotopologues or APEs, and we hope more transparent than our initial draft. We state:

“Normalizing this fractional data to glucose concentration indicated that the elevated glucose excursion in the KSPAPN mice was due to increased glucose originating from exogenous pyruvate carbons and, most substantially, non-pyruvate sources (Fig. S4E). Although this method does not specifically quantify renal gluconeogenesis, the increased contribution of non-pyruvate carbons in KSPAPN mice is reminiscent of the ¹³C dilution of TCA cycle intermediates in the kidney compared to liver, which may propagate to glucose more in KSPAPN mice than control mice. Nevertheless, we cannot rule out preexisting glucose due to impaired glucose clearance or increased gluconeogenesis from glycerol in KSPAPN mice, either of which would appear as non-pyruvate sources of glucose carbon. “

Finally, based on the reviewer’s comment, we attempted a more rigorous metabolic flux analysis (MFA) by regressing metabolite MIDs to a metabolic network model in INCA. However, as the reviewer is aware, this type of analysis assumes metabolic and isotopic steady states. Although we have done this in other projects with a pseudo steady state assumption following a labeled bolus, here the multi-compartment model required too many assumptions for our comfort level. Nevertheless, we share the results of those efforts for the reviewer’s interest.

First, we tested a series of single tissue models (either kidney or liver):

Model_01: Single tissue TCA cycle intermediates only

Model_02: Single tissue TCA cycle intermediates + Blood glucose isotopologues

Model_03: Single tissue TCA cycle intermediates only + triose isotopologues (PEP, 2PG and G3P):

Model_04: Single tissue TCA cycle intermediates only + triose isotopologues (PEP, 2PG and G3P) + Blood glucose isotopologues

Figure R3.3.1

We found that blood glucose isotopologues agreed better with liver metabolite isotopologues than kidney metabolite isotopologues (based on squared sum of residual SSR, **Figure R3.3.1**). Acceptable solutions required trioses and a source of carbon dilution at the level of glycogen or glycerol. Conversely, kidney isotopologues resulted in slightly worse SSR when the model included blood glucose isotopologues. This finding is not unexpected since the largest portion of blood glucose originates from liver, even in the setting of increased kidney adiponectin.

[FIGURE REDACTED]

Figure R3.3.2

[FIGURE REDACTED]

Figure R3.3.3

Next, we tested a dual tissue model (**Figure R3.3.2**), hoping to assign fractional contributions of glucose production between liver and kidney. The labeled pyruvate source (Pyr.tr) was fixed arbitrarily to 100. Unfortunately, this analysis

provides a range of solutions because the division of pyruvate between liver and kidney tissues cannot be determined from the data (**Figure R3.3.2**, right panels). The model tended to favor solutions where most gavaged pyruvate was metabolized in the liver to produce glucose and kidney converts a portion of pyruvate to lactate which is used in the liver compartment. Across all solutions kidney contributed to blood lactate (Figure R2). Finally, this analysis indicated that kidneys of KSPAPN mice contribute a larger fraction of blood glucose than kidneys of control mice. However, this estimate depended on the structure of the MFA model, the metabolites included in the regression, and the division of pyruvate between the tissues (**Figure R3.3.3**). Overall, while limited due to a non-steady state of experiment, this analysis is in line with other data interpretation presented in the paper. However, we note that most model solutions fell

outside of the acceptable SSR range (Figure R3.3.2, right panels) which, in combination with the complexity of the model and uncertainties of steady-state, led us to not report these findings.

• **Major 3.** Statistics: there seem to be 3 problems: (i) no correction for multiple statistical comparisons (e.g. if you test 4 timepoints x 5 substrate tolerances, by chance alone you would expect 1 timepoint to show significant difference, which is perilously close to what seen in Fig 6). (ii) no individual data points shown for some key figures, e.g. tolerance tests. (iii) when individual data is shown, sometimes it is normally distributed, in which case statistics are appropriate, and sometimes it is very far from normally distributed, e.g. 1I/J, in which case a nonparametric statistic or transformation to render the data roughly normal are needed. In such cases, often the non-parametric analysis (or parametric analysis after, e.g. log transformation) will actually show a stronger effect, because the lack of normality leads to overestimation of the relevant error, but we need to see.

Response: (i) As the reviewer suggested, we performed two-way ANOVA followed by a two-stage linear step-up procedure of Benjamini, Krieger, and Yekutieli by controlling the False Discovery Rate (<0.05) for multiple time point comparisons in the tolerance tests. (ii) We added the individual data point data for the key tolerance test (**Fig. S4**)

• **Major4.** The imaging mass spectrometry is intriguing, but some of the measured metabolites are hard to understand, e.g. what is HexCer 37:6; O5? I understand everything until the O5, but where the oxygen supposed to be? In general, there a lot of lipids reported with oxygens. Are these extra oxygens beyond those typically found in the relevant lipid? If so, what kind of structures are being proposed and how can these be validated? Also, the authors should be very clear that the observed lipids likely include fragments, e.g. DG signals may reflect TG degradation during the laser ablation.

Response: Thank you for bringing up this important point. Technically, we cannot determine the position of the “three extra oxygens” in the standard hexosylceramides, which are abbreviated as HexCer XX:Y;O2. The Lipid Map Database (<https://www.lipidmaps.org/>) does not provide the example structures for all of the species, including curated and computationally predicted species. We cannot exactly show the structure of HexCer 37:6;O5. However, among the HexCer species that we used for analysis, example structures of HexCer 42:1;O5, HexCer 43:2;O5 and HexCer 43:3;O5 are provided, which can be found at the link at the bottom of this section. HexCer 43:2;O5 and HexCer 43:3;O5 contain 6 different structures, respectively. As the reviewer pointed out, we appreciate the fact that the reported lysophospholipids or DG were generated during the ionization process (in-source fragmentation) from their larger molecule counterparts, such as TGs and phospholipids²⁰. Comparing the changes between phospholipids and lysophospholipids in adiponectin overexpression and deficient mice can provide information on the contribution of this effect to the generation of small molecule counterparts. Since the changes of these lipid species are independent of each other, the contribution of this fragmentation effect can be negligible. However, there is a possibility that significant fragmentation may occur in specific species. Therefore, we focused on the overall alterations of lipid groups rather than focusing on individual lipid species. Additionally, we provided the heatmap data that is composed of already confirmed lipid species in the kidney. To explore further, LC-MS/MS analysis can be used for validation. To clarify these technical limitations, we have added a disclosure of these limitations to the manuscript from **line 1492** as follows:

“Mass spectrometry imaging provides the levels of molecules with overlapping molecular mass. Since molecules with same molecular mass include several kinds of species, such as isobaric

species and isomeric species, further analysis by LC-MS/MS is required for the exact determination of the identity of these molecules.”

HexCer 42:1;O5: <https://lipidmaps.org/databases/lmsd/LMSP05010106>

HexCer 43:2;O5:

https://lipidmaps.org/resources/tools/chemdb_ontology?abbrev=HexCer%2043:2;O5

HexCer 43:3;O5:

https://lipidmaps.org/resources/tools/chemdb_ontology?abbrev=HexCer%2043:3;O5

- **Minor 1.** abstract line 58, it says lower utilization and accumulation of lipids. I think the authors mean lower utilization and greater accumulation?

Response: Correct! According to the reviewer’s suggestion, we corrected the phrases from “accumulation of lipids” to “greater accumulation of lipid species”.

- **Minor2.** Line 108: literature data on energy sources in different parts of the kidney are not all in alignment. Recent in vivo data from imaging mass spectrometry with isotope tracing show relatively little fat use in the cortex, more in the medulla, with the cortex heavily reliant on organic acids.

Response: Thank you so much for the important remark. We updated the reference and included the recent advancement of the spatial segregation of the metabolic role inside the kidney. The paragraph from **line 109** is as follows:

“To further complicate the situation, gluconeogenesis is energetically costly, requiring 4 ATP and 2 GTP molecules to convert pyruvate to glucose. Fatty acids and their metabolites, including acetyl coenzyme A (Acetyl CoA), citrate and ketone bodies up-regulate gluconeogenesis particularly in the renal proximal tubule ^{16,19,21}. Although fatty acids facilitate the gluconeogenesis, preference of these substrates in the kidney is spatially segregated. Glucose and free fatty acids are more prone to be utilized in the medulla, while organic acids including lactate, citrate are more consumed in the cortex ¹². Namely, renal medulla is more oxidative and glycolytic while cortex is more gluconeogenic.”

- **Minor 3.** Methods: please be more clear on how the kidney was prepared for the Seahorse analyses

Response: As the reviewer suggested, we clarified the method of the kidney tissue Seahorse-based experiments. We added the following sentences to the ***Mitochondrial respiration measurements*** section.

“Harvested kidneys were cut into half longitudinally, then punched out as a 2 mm biopsy punch (33-31, Integra Miltex). Approximately 4 mg of kidney tissues were utilized for Seahorse measurements.”

- **Minor4.** Line 897: functional data are lacking to show that altered lipid levels mediate the lower mitochondrial function and fibrotic phenotype in the KSPAKO kidneys.

Response: Thank you for bringing to our attention the missing logical connection between the observed lipid alteration and the outcomes of “fibrosis and mitochondrial dysfunction”. To clarify the possible mechanism that adiponectin mediates, we expanded upon our discussion by citing the additional references. The updated paragraph from **line 1237** is as follows:

“Although we only have a limited understanding of what each of these lipid species does, it has been demonstrated that the accumulation of glucosylceramides is associated with renal fibrosis and mitochondrial dysfunction^{22,23}. Therefore, a local adiponectin knockdown in the kidney has a profound impact on a broad set of lipids, including the accumulation of glucosylceramides, which may mediate the lower mitochondrial function and fibrotic phenotype in the KSPAKO kidneys.

• **Minor 5.** Line 957: agreed that parts of the kidney probably produce while others probably consume glucose, although many papers show that under pretty typical fasted conditions the kidney is net gluconeogenic

Response: Thank you for the important remark. To clarify the point, we have provided a more detailed discussion. The updated discussion from **line 1320** is as follows.

“Indeed, the kidney normally utilizes glucose for lactate production in the medulla, while the kidney produces glucose as a whole under fasted conditions^{11,12,24}. Thus the shift toward fat oxidation may lower glucose disposal in KSPAPN mice, causing preexisting glucose to be misinterpreted as glucose derived from non-pyruvate carbon.”

References cited above

- 1 Bjursell, M. *et al.* Opposing effects of adiponectin receptors 1 and 2 on energy metabolism. *Diabetes* **56**, 583-593, doi:10.2337/db06-1432 (2007).
- 2 Straub, L. G. & Scherer, P. E. Metabolic Messengers: Adiponectin. *Nat Metab* **1**, 334-339, doi:10.1038/s42255-019-0041-z (2019).
- 3 Holland, W. L. *et al.* Receptor-mediated activation of ceramidase activity initiates the pleiotropic actions of adiponectin. *Nat Med* **17**, 55-63, doi:10.1038/nm.2277 (2011).
- 4 Yamauchi, T. *et al.* Cloning of adiponectin receptors that mediate antidiabetic metabolic effects. *Nature* **423**, 762-769, doi:10.1038/nature01705 (2003).
- 5 Hug, C. *et al.* T-cadherin is a receptor for hexameric and high-molecular-weight forms of Acrp30/adiponectin. *Proc Natl Acad Sci U S A* **101**, 10308-10313, doi:10.1073/pnas.0403382101 (2004).
- 6 Shen, Y. Y., Charlesworth, J. A., Kelly, J. J., Loi, K. W. & Peake, P. W. Up-regulation of adiponectin, its isoforms and receptors in end-stage kidney disease. *Nephrol Dial Transplant* **22**, 171-178, doi:10.1093/ndt/gfl552 (2007).
- 7 Choi, S. R. *et al.* Adiponectin receptor agonist AdipoRon decreased ceramide, and lipotoxicity, and ameliorated diabetic nephropathy. *Metabolism* **85**, 348-360, doi:10.1016/j.metabol.2018.02.004 (2018).
- 8 Tsugawa-Shimizu, Y. *et al.* Increased vascular permeability and severe renal tubular damage after ischemia-reperfusion injury in mice lacking adiponectin or T-cadherin. *Am J Physiol Endocrinol Metab* **320**, E179-E190, doi:10.1152/ajpendo.00393.2020 (2021).
- 9 Otabe, S. *et al.* Overexpression of human adiponectin in transgenic mice results in suppression of fat accumulation and prevention of premature death by high-calorie diet. *Am J Physiol Endocrinol Metab* **293**, E210-218, doi:10.1152/ajpendo.00645.2006 (2007).
- 10 Combs, T. P. *et al.* A transgenic mouse with a deletion in the collagenous domain of adiponectin displays elevated circulating adiponectin and improved insulin sensitivity. *Endocrinology* **145**, 367-383, doi:10.1210/en.2003-1068 (2004).
- 11 Bartlett, S., Espinal, J., Janssens, P. & Ross, B. D. The influence of renal function on lactate and glucose metabolism. *Biochem J* **219**, 73-78, doi:10.1042/bj2190073 (1984).
- 12 Wang, L. *et al.* Spatially resolved isotope tracing reveals tissue metabolic activity. *Nat Methods* **19**, 223-230, doi:10.1038/s41592-021-01378-y (2022).
- 13 Dell, R. B. & Winters, R. W. Lactate gradients in the kidney of the dog. *Am J Physiol* **213**, 301-307, doi:10.1152/ajplegacy.1967.213.2.301 (1967).
- 14 Jha, M. K. & Morrison, B. M. Lactate Transporters Mediate Glia-Neuron Metabolic Crosstalk in Homeostasis and Disease. *Front Cell Neurosci* **14**, 589582, doi:10.3389/fncel.2020.589582 (2020).
- 15 Pan, X., Small, E. V., Igarashi, P. & Carroll, T. J. Generation and characterization of KspT_A and KspT_A transgenic mice. *Genesis* **51**, 430-435, doi:10.1002/dvg.22381 (2013).
- 16 Williamson, J. R., Kreisberg, R. A. & Felts, P. W. Mechanism for the stimulation of gluconeogenesis by fatty acids in perfused rat liver. *Proc Natl Acad Sci U S A* **56**, 247-254, doi:10.1073/pnas.56.1.247 (1966).

- 17 Gray, L. R. *et al.* Hepatic Mitochondrial Pyruvate Carrier 1 Is Required for Efficient Regulation of Gluconeogenesis and Whole-Body Glucose Homeostasis. *Cell Metab* **22**, 669-681, doi:10.1016/j.cmet.2015.07.027 (2015).
- 18 Georgas, K. *et al.* Use of dual section mRNA in situ hybridisation/immunohistochemistry to clarify gene expression patterns during the early stages of nephron development in the embryo and in the mature nephron of the adult mouse kidney. *Histochem Cell Biol* **130**, 927-942, doi:10.1007/s00418-008-0454-3 (2008).
- 19 Krebs, H. A., Speake, R. N. & Hems, R. Acceleration of Renal Gluconeogenesis by Ketone Bodies and Fatty Acids. *Biochem J* **94**, 712-720, doi:10.1042/bj0940712 (1965).
- 20 Hu, C., Luo, W., Xu, J. & Han, X. Recognition and Avoidance of Ion Source-Generated Artifacts in Lipidomics Analysis. *Mass Spectrom Rev* **41**, 15-31, doi:10.1002/mas.21659 (2022).
- 21 Datta, A. G. *et al.* The activation of rabbit muscle, liver, and kidney fructose bisphosphatases by histidine and citrate. *Arch Biochem Biophys* **165**, 641-645, doi:10.1016/0003-9861(74)90292-6 (1974).
- 22 Huwiler, A. & Pfeilschifter, J. Sphingolipid signaling in renal fibrosis. *Matrix Biol* **68-69**, 230-247, doi:10.1016/j.matbio.2018.01.006 (2018).
- 23 Novgorodov, S. A. *et al.* Lactosylceramide contributes to mitochondrial dysfunction in diabetes. *J Lipid Res* **57**, 546-562, doi:10.1194/jlr.M060061 (2016).
- 24 Moller, N., Rizza, R. A., Ford, G. C. & Nair, K. S. Assessment of postabsorptive renal glucose metabolism in humans with multiple glucose tracers. *Diabetes* **50**, 747-751, doi:10.2337/diabetes.50.4.747 (2001).

REVIEWER COMMENTS

Reviewer #1 (Remarks to the Author):

The authors have addressed some of the points raised in the previous review.

I am afraid, however, that previous critical concerns have been left largely unaddressed.

Although this manuscript proposes a new concept, it is riddled with contradictions that the authors should resolve appropriately before proposing a new concept.

Prior Comment 1:

In a further analysis of AdipoR2 KO mice, the authors demonstrated an increase in gluconeogenesis in the kidney during the glutamine tolerance test. However, this contradicts the results of other experiments in the original manuscript, where they claim that adiponectin enhances gluconeogenesis in the kidney. On one hand, they should state in this manuscript that adiponectin may inhibit gluconeogenesis via AdipoR2 in the kidney. On the other hand, if they assume that the AdipoR2 KO mice showed increased gluconeogenesis in the kidney just because of an indirect effect of AdipoR2 deficiency in other tissues than kidney, they should generate and analyze kidney-specific AdipoR2 KO mice.

Prior Comment 3:

The authors have not answered the previous question. They have previously shown that blood glucose levels improve in mice that overexpress adiponectin in adipose tissue (Endocrinology 145, 367-383, 2004). They should verify whether these mice would show increased gluconeogenesis in the kidney. If these mice do not exhibit increased gluconeogenesis in the kidney, they should demonstrate that circulating adiponectin and intracellularly expressed adiponectin have distinct mechanisms of action on gluconeogenesis.

Reviewer #2 (Remarks to the Author):

The authors have adequately addressed my concerns and I believe the paper is improved like this

Reviewer #3 (Remarks to the Author):

This paper contains a lot of interesting data, even more than before, and is suitable for publication with some final minor clarifications regarding the isotope tracing analyses (which are nicely improved):

The figures refer accurately to product metabolites from pyruvate (lactate, glucose) from 3 sources:

- Endogenous pyruvate
- Exogenous pyruvate
- Other

I agree that the data generally support these categories (although for glucose, which is more slowly turning over, some of the "other" could be lack of steady-state labeling).

I further agree with the text lines 754-758 defining total FC and the FC specific to exogenous pyruvate. But there seems to be an error in the subsequent text (Not sure if this error was propagated into the figures): "The difference between these values 759 represents the FC from sources that do not enter the pyruvate pool." Instead, I believe the correct meaning is "The difference between these values represents the FC from endogenous pyruvate" (i.e. $FC_{\text{endogenous}} = FC_{\text{total}} - FC_{\text{exogenous}}$) and the Difference between the FC_{total} and 1 represents the FC from sources that do not enter the pyruvate pool ($FC_{\text{other}} = 1 - FC_{\text{total}}$)?

Replies to Reviewer 1 remaining comments

We thank Reviewer1 for the additional thoughtful suggestions and further careful assessment of our manuscript. We respond to the important question as to what distinguishes the roles of circulating vs. intracellular adiponectin. We have performed glutamine tolerance tests in our systemic Δ Gly adiponectin overexpression mouse model and revised the discussion to clarify and address the concerns. Major changes were highlighted in yellow in the revised manuscript.

Reviewer #1 (Remarks to the Author):

The authors have addressed some of the points raised in the previous review. I am afraid, however, that previous critical concerns have been left largely unaddressed. Although this manuscript proposes a new concept, it is riddled with contradictions that the authors should resolve appropriately before proposing a new concept.

Prior Comment 1:

In a further analysis of AdipoR2 KO mice, the authors demonstrated an increase in gluconeogenesis in the kidney during the glutamine tolerance test. However, this contradicts the results of other experiments in the original manuscript, where they claim that adiponectin enhances gluconeogenesis in the kidney. On one hand, they should state in this manuscript that adiponectin may inhibit gluconeogenesis via AdipoR2 in the kidney. On the other hand, if they assume that the AdipoR2 KO mice showed increased gluconeogenesis in the kidney just because of an indirect effect of AdipoR2 deficiency in other tissues than kidney, they should generate and analyze kidney-specific AdipoR2 KO mice.

Response: Thank you for this important remark. We are addressing the gluconeogenic function of renal *intracellular* adiponectin, not the *circulating* adiponectin. In the kidneys of adiponectin overexpressing mice, the gluconeogenic function is enhanced, with an increase in both systemic circulating and endogenous kidney adiponectin. To differentiate from the effects of circulating adiponectin, we utilized kidney specific adiponectin KO mice that enable us to deplete renal adiponectin selectively, without affecting the circulating adiponectin levels. The effects are quite clear.

In this revision, we report detecting a higher level of glucose in adipoR2 KO mice during a glutamine tolerance test, which suggests a functional involvement of adipoR2 in *suppressing* renal gluconeogenesis. This result clearly highlights a dichotomy between *circulating* adiponectin and *renal intracellular* adiponectin. Based on these points, we revised the manuscript from line 708 as follows:

“Given that glutamine is the substrate preferentially utilized by the kidney rather than by the liver, renal AdipoR2 seems to suppress gluconeogenesis in the kidney, similar to its actions in the liver. Additionally, we assessed the renal gluconeogenesis by utilizing Δ Gly adiponectin overexpressing mice. This mouse displays an increased circulating level of adiponectin in plasma, derived from adipocytes rather than the kidney. The adiponectin overexpressing mice exhibited a very diminished response in a glutamine tolerance test (Fig. S5D). This suggests that renal intracellular adiponectin rather than circulating adiponectin is the driver for enhanced gluconeogenesis in the kidney.”

Prior Comment 3:

The authors have not answered the previous question. They have previously shown that blood glucose levels improve in mice that overexpress adiponectin in adipose tissue (Endocrinology 145, 367-383, 2004). They should verify whether these mice would show increased gluconeogenesis in the kidney. If these mice do not exhibit increased gluconeogenesis in the kidney, they should demonstrate that circulating adiponectin and intracellularly expressed adiponectin have distinct mechanisms of action on gluconeogenesis.

Response: Again, an excellent point that can be better understood in the context of *Prior Comment 1*. As the reviewer suggested, we performed a glutamine tolerance test with the Δ Gly adiponectin overexpression mice. As shown in **Fig. S5D**, Δ Gly adiponectin overexpression displayed a minimal impact on glucose level during glutamine tolerance test. This data suggest adipose tissue derived circulating adiponectin does not enhance gluconeogenesis in the kidney. As the reviewer points out, this result clarifies the distinct roles between the *circulating* and *intracellular* adiponectin. To underscore this point, we added the discussion from line 1078 as follows:

“In terms of renal gluconeogenesis, AdipoR2 may suppress the renal gluconeogenetic pathway, since the glucose levels during a glutamine tolerance test were higher in AdipoR2 KO mice, even though the impact of AdipoR2 deficiency on renal gluconeogenic gene expression was limited. To analyze the impact of adipocyte derived circulating adiponectin (i.e. not derived from the kidney), we also assessed the renal gluconeogenesis in Δ Gly adiponectin overexpressing mice¹. Higher amounts of adipocyte-derived circulating adiponectin in the Δ Gly mice had a minimal impact on renal gluconeogenesis, highlighting the importance of the source of adiponectin to manifest its effects on gluconeogenesis. Given that the AdipoRs are mainly localized in the plasma membrane², these data highlight the distinct roles of circulating vs. intracellular adiponectin.”

Literature cited:

- 1 Combs, T. P. *et al.* A transgenic mouse with a deletion in the collagenous domain of adiponectin displays elevated circulating adiponectin and improved insulin sensitivity. *Endocrinology* **145**, 367-383 (2004). <https://doi.org:10.1210/en.2003-1068>
- 2 Yamauchi, T. *et al.* Cloning of adiponectin receptors that mediate antidiabetic metabolic effects. *Nature* **423**, 762-769 (2003). <https://doi.org:10.1038/nature01705>